# Zyxin regulates endothelial von Willebrand factor secretion by reorganizing actin filaments around exocytic granules

Xiaofan Han[1,*], Pin Li[1,*], Zhenghao Yang[1], Xiaoshuai Huang[2], Guoqin Wei[1], Yujie Sun[3], Xuya Kang[1], Xueting Hu[1], Qiuping Deng[1], Liangyi Chen[2], Aibin He[2], Yingqing Huo[1], Dong Li[4,5], Eric Betzig[4] & Jincai Luo[1]

Endothelial exocytosis of Weibel–Palade body (WPB) is one of the first lines of defence against vascular injury. However, the mechanisms that control WPB exocytosis in the final stages (including the docking, priming and fusion of granules) are poorly understood. Here we show that the focal adhesion protein zyxin is crucial in this process. Zyxin downregulation inhibits the secretion of von Willebrand factor (VWF), the most abundant cargo in WPBs, from human primary endothelial cells (ECs) induced by cAMP agonists. Zyxin-deficient mice exhibit impaired epinephrine-stimulated VWF release, prolonged bleeding time and thrombosis, largely due to defective endothelial secretion of VWF. Using live-cell super-resolution microscopy, we visualize previously unappreciated reorganization of pre-existing actin filaments around WPBs before fusion, dependent on zyxin and an interaction with the actin crosslinker α-actinin. Our findings identify zyxin as a physiological regulator of endothelial exocytosis through reorganizing local actin network in the final stage of exocytosis.

[1] Laboratory of Vascular Biology, Institute of Molecular Medicine, Beijing Key Laboratory of Cardiometabolic Molecular Medicine, Peking University, Beijing 100871, China. [2] The State Key Laboratory of Biomembrane and Membrane Biotechnology, Beijing Key Laboratory of Cardiometabolic Molecular Medicine, Laboratory of Institute of Molecular Medicine, Peking University, Beijing 100871, China. [3] State Key Laboratory of Biomembrane and Membrane Biotechnology, Biodynamic Optical Imaging Center (BIOPIC), School of Life Sciences, Peking University, Beijing 100871, China. [4] Janelia Research Campus, Howard Hughes Medical Institute, Ashburn, Virginia 20147, USA. [5] National Laboratory of Biomacromolecules, Institute of Biophysics, Chinese Academy of Sciences, Beijing, 100101, China. * These authors contributed equally to this work. Correspondence and requests for materials should be addressed to D.L. (email: lidong@ibp.ac.cn) or to J.L. (email: jincailuo@pku.edu.cn).

Endothelial cells (ECs) are equipped with specific secretory granules (SGs) known as Weibel–Palade bodies[1] (WPBs) in which various bioactive molecules are stored and rapidly released upon stimulation[2]. The most abundant cargo in WPBs is von Willebrand factor (VWF), a multimeric glycoprotein that is required for WPB formation. When vascular injury occurs, VWF is secreted locally following cellular activation where it participates in developing haemostatic plug or thrombus. Impaired production and secretion of VWF are known causes of von Willebrand disease (VWD), the most common inherited bleeding disorder worldwide[3]. Endothelial WPB exocytosis can be induced by both $Ca^{2+}$ and cAMP-elevating agonists. The cAMP-mediated secretion from ECs is a fundamental process in the physiological regulation of plasma VWF level, which is raised in response to epinephrine, a hormone released in physiological conditions[4]. The importance of this pathway is further highlighted by the rise in VWF levels in patients with VWD and mild hemophilia A following administration of epinephrine or the vasopressin analogue DDAVP[5]. However, the mechanisms of WPB exocytosis in the final stage are poorly understood.

Actin plays important roles in regulated exocytosis, a cellular process fundamental to normal physiology[6–11]. A dual role for cortical actin in controlling exocytosis has been proposed[6,7,9,10]. In addition to a well-described negative regulator that behaves as a physical barrier to prevent granules' access to the cell surface[6,7], actin filaments act as an enhancer of exocytosis by providing tracks for the translocation of SGs[11], or forming a coat around SGs to promote cargo release[9,10,12,13]. The actin coat formation plays a particular role in promoting exocytosis of the large SGs such as glue granules in *Drosophila* salivary gland[13] and WPBs in ECs[14]. It has been suggested that, upon stimulation, cortical actin is first cleared to allow SGs for fusion[10,12], which triggers actin coat formation via a *de novo* nucleation process[10,15]. Cortical actin remodelling is regulated by the signalling molecules in the focal adhesions (FAs)[16], which appear to be the microdomains of exocytosis[17]. Zyxin is an important FA protein with three LIM domains at its C-terminus, which are essential for its localization to FA. Zyxin regulates actin remodelling through the interaction with multiple actin regulatory binding partners[18]. Nevertheless, it is unknown whether Zyxin-coupled actin remodelling is involved in regulation of exocytosis.

Here we show that FA protein zyxin plays an essential role in the *in vitro* and *in vivo* release of VWF, by mediating the formation of actin frameworks around exocytic WPBs from the pre-existing cortical actin network before fusion.

## Results

**Zyxin is a regulator of cAMP-mediated VWF secretion.** FAs appear to be exocytostic hotspots[19]. We now show that WPBs are aligned along actin fibres anchored to FAs (Supplementary Fig. 1). Together, these findings suggest a possible role of FA components in WPB exocytosis. We therefore performed a short-hairpin RNA (shRNA) screen targeting common FA genes[19] to assess their effects on the VWF secretion from human umbilical vein ECs (HUVECs) induced by a cAMP-elevating agonist forskolin. The exocytotic roles of several previously identified genes, including Rac1 (ref. 20) and myosin IIA[14], were confirmed (Fig. 1a). In addition, however, the secretion of VWF is mostly suppressed by the shRNA of zyxin and validated by two distinct shRNAs (shZyxin-1 and -2) (Fig. 1b). These two SHRNAs displayed similar knockdown efficiency on zyxin expression, and thus shZyxin-1 (abbreviated as shZyxin) was used for the subsequent experiments. To visualize the effect of zyxin on WPB exocytosis, shZyxin was

specifically expressed in DsRed-labelled cells and WPBs were assessed by VWF immunostaining (Supplementary Fig. 2a). While there was no significant difference in the number of WPBs between shZyxin-expressing cells and control cells under quiescent conditions ($P < 0.05$, Student's $t$-test), the number of WPBs in shZyxin-expressing cells decreased less dramatically than that in control cells with forskolin stimulation (Supplementary Fig. 2b). Zyxin is located at FAs and along stress fibres[18]. To determine the specificity of zyxin-mediated effects, we tested three additional reagents: another intracellular cAMP-elevating agonist (epinephrine), and two $Ca^{2+}$-raising agonists (thrombin and histamine). Zyxin knockdown reduced the epinephrine-induced VWF secretion, similar to that induced by forskolin (Fig. 1c). However, zyxin knockdown did not significantly affect VWF secretion in response to the two $Ca^{2+}$-raising agonists (Fig. 1d). These results suggest that zyxin preferentially regulates cAMP signalling-mediated VWF secretion.

**Pre-existing filaments were assembled around WPBs.** Considering the involvement of zyxin in actin remodelling, it was tempting to speculate that zyxin regulates WPB exocytosis via actin remodelling. To simultaneously visualize the dynamics of fine cortical actin filaments and the behaviour of exocytic granules in close proximity to the plasma membrane at spatial resolution beyond diffraction limit, we used ultrahigh numerical aperture total internal reflection fluorescence structured illumination microscopy (high NA TIRF-SIM)[21]. This system is able to image intracellular dynamics at 84-nm resolution for up to 200 frames at subsecond acquisition times while using an order of magnitude lower illumination intensities than other super-resolution techniques[21], capturing fine spatiotemporal details over the long exocytosis process. We co-expressed mCherry or GFP-Lifeact (an actin-binding peptide to monitor the behaviour of actin filaments[22]) and GFP-VWF or mCherry-PSL-lum (a fusion protein of the luminal part of P-selectin with mCherry at the C-terminus) in HUVECs, as these combinations are well characterized for imaging actin structures and WPBs simultaneously[14]. While the exocytic behaviour of mCherry-PSL-lum before fusion was similar to that of GFP-VWF (Supplementary Fig. 3a, Supplementary Movie 1) in HUVECs induced by forskolin, we found that the combination of mCherry-PSL-lum and GFP-Lifeact (Supplementary Movies 2 and 3) displayed better imaging quality than that of GFP-VWF and mCherry-Lifeact (Supplementary Movies 4 and 5) and thus was used for most subsequent experiments. In cells co-expressing mCherry-PSL-lum and GFP-Lifeact, ring-like actin structures around exocytic WPBs were easily detected by TIRF-SIM in the presence of forskolin (Fig. 2a,b and Supplementary Movie 2). In a previous study, the actin coat formation on WPBs was indicated as a postfusion *de novo* nucleation process of actin filaments[14]. Interestingly, TIRF-SIM images showed that actin frameworks, which were formed from the pre-existing filaments, were observed in 80% ($n = 77$) of successfully exocytosed WPBs before fusion (Fig. 2c). Such actin frameworks were also observed in HUVECs induced by epinephrine (Supplementary Fig. 3b) and phorbol-12-myristate-13-acetate (Supplementary Fig. 3c), an agonist used in a previous study of WPBs[14].

The actin framework was formed in a mean time of $7 \pm 3$ s ($n = 77$; Fig. 2a,b), which usually started from a nearby actin bundle that was relatively thick. The bundle then curved and merged with neighbouring actin filaments to assemble as an actin framework (Fig. 2a, Supplementary Movie and 3). In response to stimulation, there is a morphological change of WPBs from a rod shaped to a spherical structure, which occurs upon fusion[23]. Notably, the assembly of the pre-existing filaments started when

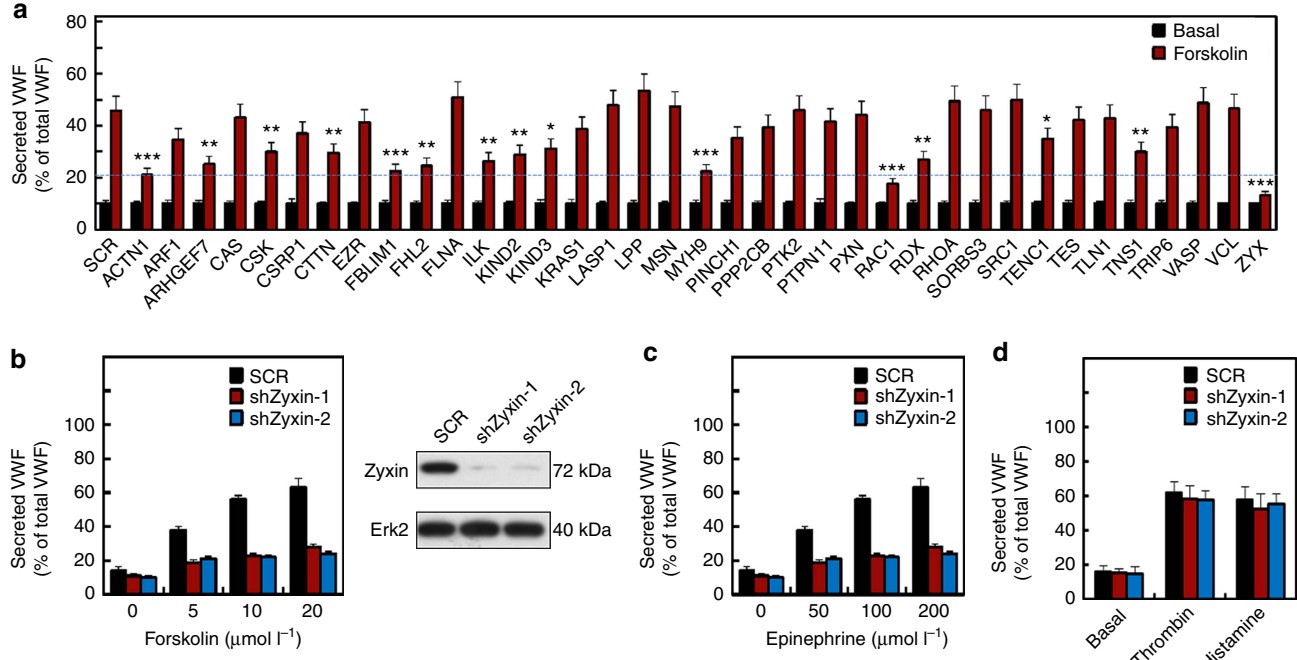

**Figure 1 | Zyxin is required for WPB exocytosis mediated via cAMP pathway. (a)** ELISA of VWF secreted from HUVECs expressing scrambled shRNA (SCR) or shRNAs targeting 37 FA genes, with forskolin stimulation. The bar graph shows the ratio of secreted VWF to cytoplasmic VWF ($n = 12$, *$P < 0.05$, **$P < 0.01$, ***$P < 0.001$, versus scrambled shRNA, 3 independent experiments). **(b)** ELISA of VWF secreted from HUVECs expressing scrambled (SCR) or zyxin shRNAs (shZyxin-1 and -2) with stimulation of forskolin (left panel) ($n = 15$, ***$P < 0.001$, 3 independent experiments). The efficacy of zyxin knockdown was confirmed by western blotting analysis using an anti-zyxin antibody with ERK2 as a loading control (right panel). **(c,d)** ELISA of VWF secreted from HUVECs expressing scrambled (SCR) or zyxin shRNAs (shZyxin-1 and -2) with epinephrine **(c)** or thrombin (1 U ml$^{-1}$) and histamine (10 μmol l$^{-1}$) stimulation at the indicated concentration **(d)**. ($n = 4$, ***$P < 0.001$). All error bars represent s.d. Student's $t$-test was used for statistical analysis. The uncropped scans of western blottings were shown in Supplementary Fig. 9.

the WPBs were still rod shaped (Fig. 2a), indicating that the formation of actin framework is a prefusion event. We observed these same phenomena in HUVECs expressing GFP-VWF and mCherry-Lifeact (Supplementary Fig. 4, Supplementary Movies 4 and 5).

To examine whether new actin is recruited from the cytoplasm to sites of the existing filament framework around the exocytic vesicle, we performed the fluorescent G-actin incorporation assay[24]. The signal of Alexa Fluor 647–G-actin was hardly detected on the reorganized pre-existing filaments before fusion, suggesting that the reorganization of pre-existing filaments contributes to the formation of actin framework. G-actin was recruited to form the actin coat structure on WPBs upon fusion within the frameworks (11 out of the 15 across 4 cells; Fig. 2d,e).

Interestingly, we found a significant decline in the fluorescence intensity once the WPBs became spherical, followed by expulsion of VWF (Fig. 2a and Supplementary Fig. 4). This suggests a retraction of WPBs into the cytoplasm, likely due to the contraction of actin coat to facilitate full release of VWF. Further evidence was provided by three-dimensional image analysis using a variable-angle TIRF[25] (Supplementary Fig. 5, $n = 6$), although the effect of plasma membrane deformation cannot be excluded.

**Zyxin mediates the formation of actin framework.** We then investigated whether zyxin affects the formation of actin framework around exocytic WPBs. The number of the actin framework formation around exocytic WPBs in each shZyxin-expressing cell ($2 \pm 1$, $n = 6$) was rare compared with that in

the control cells ($10 \pm 3$, $n = 7$, $P = 0.0064$, Student's $t$-test; Fig. 2f,g and Supplementary Movie 6). Furthermore, the rare actin frameworks in shZyxin-expressing cells were dysfunctional and cannot expel the cargo efficiently (Fig. 2g, and Supplementary Movie 7). Thus zyxin plays a key role for the formation and maintenance of actin framework with structural and functional integrity. To directly visualize the involvement of zyxin in the formation of actin framework during WPB exocytosis, we used cells co-expressing Halo-zyxin, which was detected by a fluorescent JF$_{549}$-HaloTag ligand[26], and GFP-Lifeact or GFP-VWF, respectively. TIRF-SIM images showed that, with forskolin stimulation, zyxin was enriched around 54% of the exocytic WPBs at a mean time of $12 \pm 4$ s ($n = 53$) before they fused with the plasma membrane (Fig. 3a,b and Supplementary Movie 8), while zyxin was co-localized with actin frameworks in 23 of the 32 (71%) observed events (Fig. 3c,d and Supplementary Movie 9). In addition, triple immunostaining showed that zyxin was co-localized with the actin framework in 32 of the 47 (68%) exocytic WPBs in fixed cells with forskolin stimulation (Fig. 3e,f). Collectively, these data suggest that zyxin mediates the formation of the actin framework on exocytic WPBs.

**Phosphorylation of zyxin is critical to VWF secretion.** Having established the role of zyxin in actin framework formation, we further dissected the underlying molecular mechanism. The function of zyxin is regulated by the phosphorylation of several residues, including serines 142 and 143 (S142/143)[27]. We found that both forskolin and epinephrine rapidly induced S142/143 phosphorylation (Fig. 4a and Supplementary Fig. 6a). Unlike

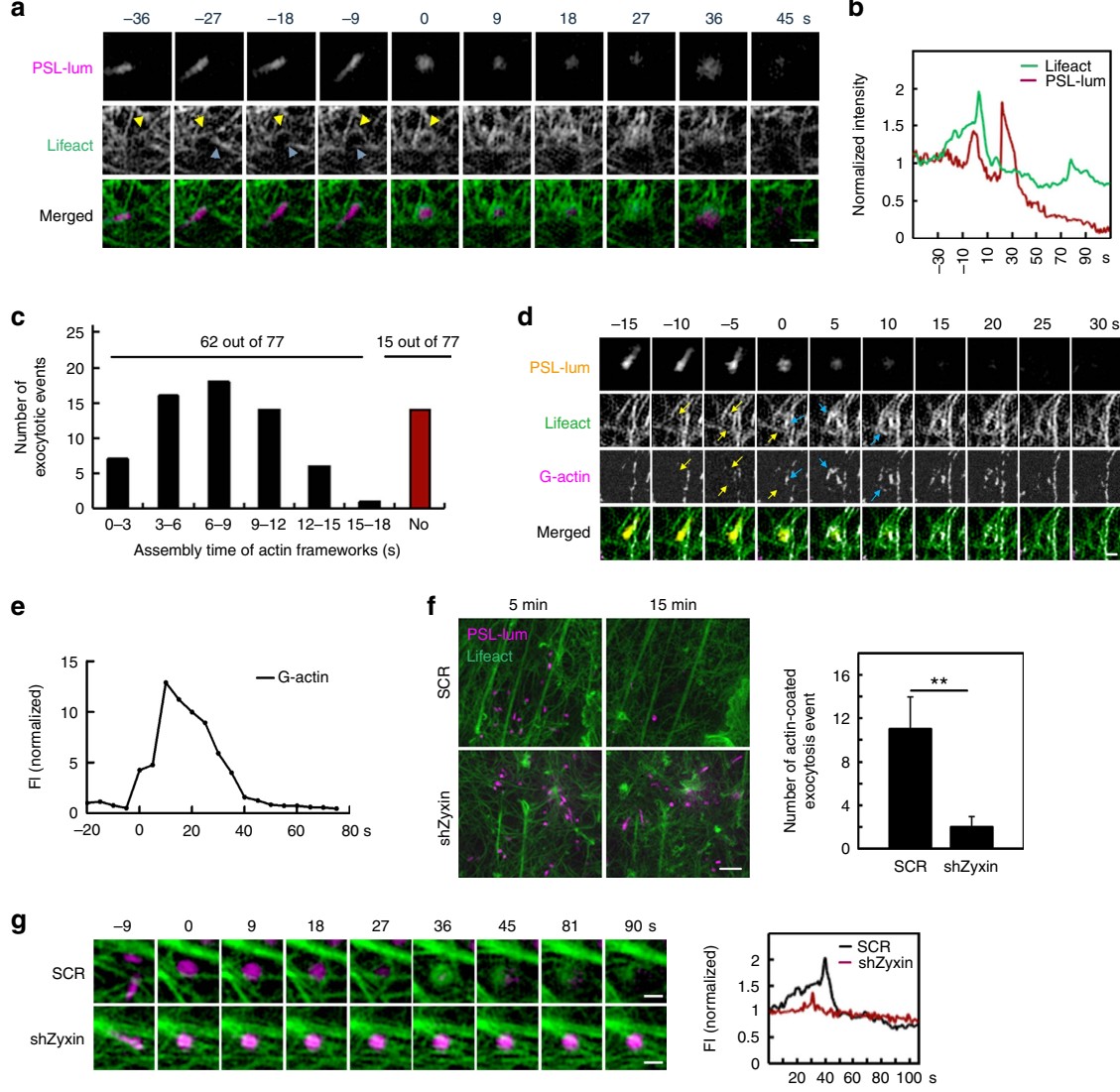

**Figure 2 | Zyxin is essential to the formation of actin framework from pre-existing filaments to promote WPB exocytosis.** (**a**) Time-lapse series showing the exocytosis of an individual WPB in a HUVEC co-expressing a fusion protein, mCherry-PSL-lum (the luminal part of P-selectin with mCherry at the C-terminus, magenta) and GFP-Lifeact (green). (**b**) The normalized fluorescence time trace of GFP-Lifeact and mCherry-PSL-lum. (**c**) Assembling time for pre-existing actin filaments to form actin framework ($n = 77$, across 10 cells, 3 independent experiments). (**d**) Left: time-lapse series showing the incorporation of G-actin, labelled by perfusing the saponin/AF647-g-actin solution, containing forskolin/IBMX at the same time. The yellow arrows indicate the pre-existing filaments formed by polymerization. The blue arrows indicate the newly recruited actin. Right: the quantified fluorescence time trace of AF647-G-actin. Time 0 is defined as the fusion point. ($n = 4$). (**e**) HUVECs co-expressing mCherry-PSL-lum (magenta) and GFP-Lifeact (green) were infected with scrambled shRNA (SCR) or shZyxin (the abbreviation of shZyxin-1, which is used thereafter). Images were acquired every 3 s from 5 min to 15 min after forskolin stimulation. The still images show the first (5 min) and last (15 min) imaging time points. (**f**) The average number of actin framework around exocytic WPBs in each SCR and shZyxin cell ($n = 8$, \*\*$P < 0.01$, 3 independent experiments). (**g**) Left: a representative process of actin framework formation in SCR and shZyxin cells. Right: the fluorescence time trace of F-actin around the WPBs. Scale bars, 500 nm in panels (**a**,**g**); 2 μm in (**f**); 1 μm in (**d**). All error bars represent s.d. Student's *t*-test was used for statistical analysis.

wild-type (WT) zyxin, a zyxin S142/143A mutant in which the S142/143 sites were changed to alanines to abolish the phosphorylation did not significantly rescue the defect in shZyxin-expressing cells (Fig. 4b), suggesting that S142/143 phosphorylation is essential for zyxin-mediated VWF secretion. Protein kinase A (PKA) and exchange protein directly activated by cAMP (EPAC) are both effectors of cAMP signalling[20]. S142/143 phosphorylation was inhibited by treatment with a PKA-specific inhibitor but not an EPAC inhibitor (Brefeldin A), suggesting that zyxin functions downstream of PKA (Fig. 4c, Supplementary Fig. 6b,c).

**Zyxin–α-actinin interaction is required for VWF secretion.** Vasodilator-stimulated phosphoprotein (VASP) and α-actinin are important binding partners of zyxin function[18]. Interestingly, α-actinin knockdown significantly decreased the forskolin-induced VWF secretion but VASP knockdown did not (Supplementary Fig. 6d,e). Furthermore, using two zyxin mutants[18] deficient in binding α-actinin or VASP, respectively, we found that introduction of the α-actinin-binding mutant, but not the VASP-binding mutant, failed to rescue the actin framework formation and WPB exocytosis in shZyxin-expressing cells ($n = 4$; Fig. 5a,b). Indeed, forskolin induced an interaction

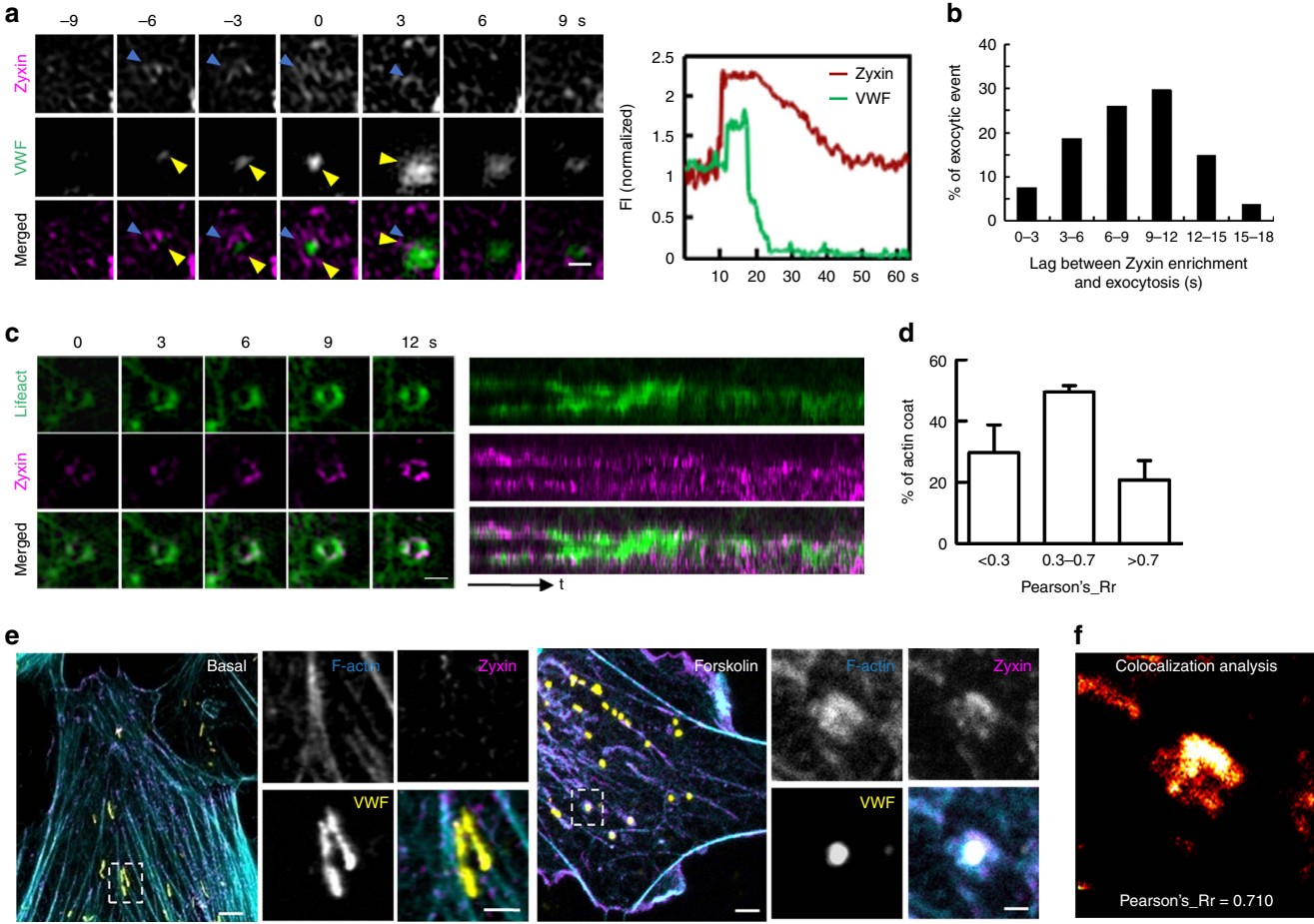

**Figure 3 | Zyxin is enriched on actin framework around exocytic WPBs. (a)** Left: TIRF-SIM image of a HUVEC co-expressing Halo-zyxin (magenta) and GFP-VWF (green). Right: the quantified fluorescence time trace of GFP-VWF and Halo-zyxin ($n = 53$, across 8 cells, 3 independent experiments). The blue arrows indicate the enriched zyxin and the yellow arrows indicate the exocytic WPB. **(b)** The lag time between zyxin enrichment around the exocytic WPB and the fusion point, based on 53 exocytosis events across 8 cells (26 of the events had no evident zyxin enrichment, 3 independent experiments). **(c)** Left: the still image of a HUVEC co-expressing Halo-zyxin (magenta) and GFP-lifeact (green). Right: the fluorescence kymograph tracing of Halo-zyxin and GFP-Lifeact. **(d)** The statistical analysis of the colocalization of zyxin and actin coat of **(c)** with the Pearson's coefficient. ($n = 32$, across 6 cells, 2 independent experiments). **(e)** Confocal images of immunofluorescent staining of HUVECs under basal conditions (Basal) or forskolin stimulation (Forskolin) with Phalloidin (cyan) and the antibody against VWF (yellow) or zyxin (magenta). The images on the right panels correspond to the magnified insets; dashed lines represent regions in which $y$–$z$ sections were taken. **(f)** Colocalization analysis of actin and zyxin in **(e)**. The Pearson's coefficient is analysed by ImageJ. All error bars represent s.d. Student's $t$-test was used for statistical analysis. Scale bars, 500 nm in panels (**a**,**c**) and magnified insets of (**e**), 1 μm in (widefields of **e**).

between zyxin and α-actinin, which was dependent on S142/143 (Fig. 5c). Triple immunostaining showed a distinct accumulation of α-actinin in the actin frameworks surrounding the WPB, which was absent in shZyxin-expressing cells (Fig. 5d). Together, these results show that α-actinin is required for the formation of actin frameworks on secretory WPBs and therefore plays a crucial role in the regulation of VWF secretion in the zyxin-mediated pathway.

**Zyxin deletion impairs vascular haemostasis and thrombosis.** To understand the function of zyxin in regulating exocytosis under physiological conditions, we compared the plasma VWF levels in WT and zyxin knockout (Zyxin KO) mice[27,28] before and after intraperitoneal (i.p.) injection of epinephrine. There was no significant difference in the plasma VWF levels of WT and Zyxin KO mice before epinephrine injection. In contrast, the plasma VWF levels were significantly increased by epinephrine injection in WT mice but not in Zyxin KO mice

(Fig. 6b). We then investigated the consequences of the defective response in plasma VWF levels to epinephrine in Zyxin KO mice, using a tail bleeding time and a mesenteric FeCl$_3$-induced thrombus-formation assay. The bleeding time in Zyxin KO mice was significantly longer than in WT mice (Fig. 6c). Similarly, compared with WT mice, the thrombus formation in Zyxin KO mice was impaired, as shown by rhodamine-labelled platelets (Fig. 6d).

To examine whether the expression of zyxinin blood cells (BC), such as platelets, monocytes and leucocytes, affects the plasma level of VWF and its related haemostasis and thrombosis, we carried out cross bone marrow transplantations. Zyxin KO bone marrow cells were transplanted in the WT recipient mice, establishing the Zyxin EC mice model (EC$^{Zyx+}$/BC$^{Zyx-}$). Meanwhile, WT bone marrow cells were transplanted into the Zyxin KO recipient to establish the Zyxin BC model (EC$^{Zyx-}$/BC$^{Zyx+}$) (Fig. 6a, Supplementary Fig. 7). The plasma VWF levels were significantly increased by epinephrine in Zyxin EC mice, while the VWF level in Zyxin BC mice was

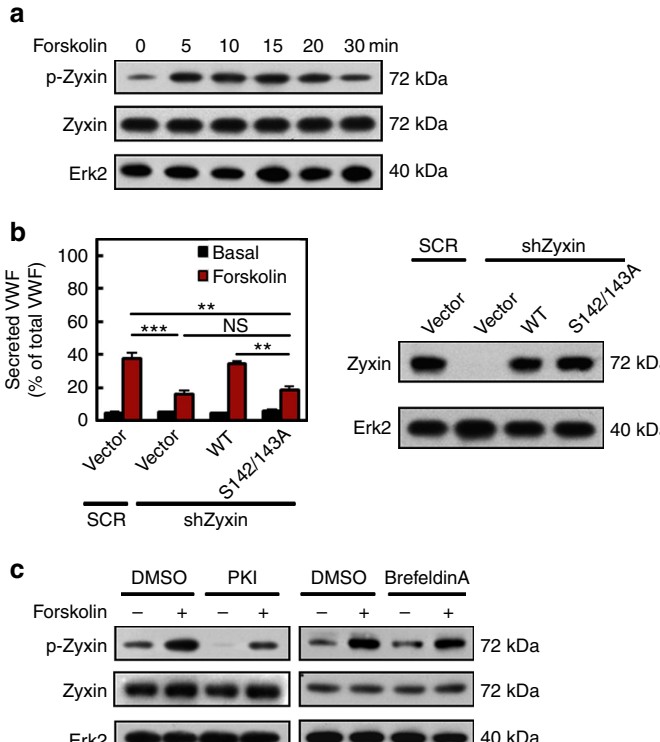

**Figure 4 | Zyxin-mediated VWF secretion depends on the PKA activity.**
(**a**) Western blotting of phospho-zyxin (p-Zyxin) in HUVECs stimulated
with forskolin for different time points. (**b**) Left: ELISA of VWF secreted from
SCR HUVECs and shZyxin HUVECs that was rescued with empty vector,
zyxin or the zyxin S142/143A mutant, respectively, with forskolin
stimulation ($n=13$, **$P<0.01$, ***$P<0.001$, NS $P>0.05$, 3 independent
experiments). Right: western blotting analysis using an anti-zyxin antibody
that recognized both zyxin and the zyxin-S142/143A mutant. (**c**) Western
blotting of phospho-zyxin S142/143 in HUVECs stimulated with forskolin in
the presence of specific inhibitor of PKA (PKI) or EPAC (Brefeldin A).
All error bars represent s.d. Student's $t$-test was used for statistical analysis.
The uncropped scans of western blottings were shown in Supplementary
Fig. 9.

slightly higher ($P=0.048$) than that in Zyxin KO mice (Fig. 6b).
This result suggests that Zyxin expression in ECs is critical for
maintaining plasma VWF level under stress. Consistently, the tail
bleeding time of Zyxin BC mice was not significantly shorter than
that of Zyxin EC mice (Fig. 6c). Similarly, the thrombus
formation was only slightly corrected in Zyxin BC compared
with Zyxin KO mice (Fig. 6d). In contrast, the thrombus
formation in Zyxin EC mice was only slightly impaired compared
with WT mice (Fig. 6d). These data suggest that zyxin-
mediatedVWF release from ECs play a major role in maintaining
haemostasis and thrombosis under stress. This result is also
consistent with a previous observation that VWF is mainly
synthesized in ECs and endothelial VWF predominantly
contributes to normal haemostasis and thrombosis[29].

**Zyxin-deleted ECs exhibit defective WPB exocytosis.** To directly
assess the role of zyxin in regulation of VWF secretion
from mouse ECs, we further compared epinephrine-induced
VWF secretion and the WPB-associated actin framework of
ECs isolated from the hearts of WT and Zyxin KO mice (Fig. 7a).
Upon stimulation, the number of α-actinin-decorated actin
frameworks around WPBs in ECs from WT mice was sig-
nificantly higher than that from Zyxin KO mice ($n=21$;

Fig. 7b,c). Consistent with this, after stimulation, the number
of WPBs remaining in the cytoplasm of ECs from the KO mice
was significantly higher than that in ECs from WTs (Fig. 7c).
These data confirm that zyxin deficiency impairs epinephrine-
induced endothelial VWF secretion. Taken together, the above
results demonstrate that zyxin-mediated endothelial exocytosis
of WPBs is essential for stress-induced haemostasis.

## Discussion
In this study, by using live-cell super-resolution microscopy to
simultaneously visualize the spatiotemporal dynamics of
fine cortical actin filaments and the behaviour of exocytosing
WPBs in close proximity to the plasma membrane, we reveal
previously unappreciated reorganization of pre-existing actin
filaments around WPBs before fusion, which remain there until
cargo release is complete (Fig. 2a). This is in contrast to
a previous study with confocal microscopy under which
WPB exocytosis-coupled fine actin filaments only appeared to
be a diffuse haze before fusion[14] (Supplementary Fig. 8). Such
actin frameworks were observed in 80% ($n=77$) of successfully
exocytosed WPBs before fusion. Interestingly, G-actin was
recruited to form a ring structure (actin coating) on WPBs
upon fusion within the frameworks (Fig. 2d). These findings
suggest a synergetic strategy for cortical actin network in effective
regulation of WPB exocytosis (Fig. 8). Upon stimulation, the
pre-existing actin filaments dynamically interact with exocytic
granules to form the actin frameworks, which limit granule
movement and promote granules close to the plasma membrane.
Subsequently, actin monomers are recruited from the cytosol
and form coat structures around granules within the actin
frameworks upon fusion. As such, cells coordinate the dynamic
changes of actin filaments around exocytic granules to effectively
and precisely control cargo release.

Emerging evidence suggests that actin filaments within
the cortex, which are derived from either the remaining or
newly forming actin filaments, are present before fusion due to
partial disassembly of actin network but cannot be clearly
visualized by conventional light microscopy[30–32]. For example, in
activated natural killer cells, cortical actin appeared to be cleared
under conventional light microscopy but can be seen under
super-resolution microscopy[31,32]. Importantly, these cells require
some degree of actin filaments present for agonist-induced
secretion. Therefore, the indispensable role of such under-
appreciated cortical actin network may represent a general
paradigm that is employed in the final stages of exocytosis. It is
also worth noting that, in addition to WPBs, secretory vesicles in
other cell types, such as neurosecretory cells[11] and melanocytes[33],
need to undergo translocation from the cytosol to the plasma
membrane, which was facilitated by remodelling of cortical
actin network. However, the molecular mechanism underlying
the interactions between the exocytotic granules and pre-existing
actin filaments largely remain unknown. In this study, we
have shown that zyxin mediates the formation of actin
frameworks on exocyticWPBs, involving α-actinin. Further
study is needed to dissect the detailed mechanism involved in
this process.

While zyxin is known for its mechanosensing function in
the maintenance of actin fibre integrity[18,27], we have now
demonstrated that zyxin is an important regulator of cAMP
agonist-induced WPB exocytosis in cultured primary human
ECs and mouse models of haemostasis and thrombosis.
Mechanistically, PKA-dependent zyxin phosphorylation leads to
the recruitmentof α-actinin, hence mediating actin framework
formation in promotion of WPB exocytosis. The importance of
cAMP-mediated secretion has been highlighted by its clinical

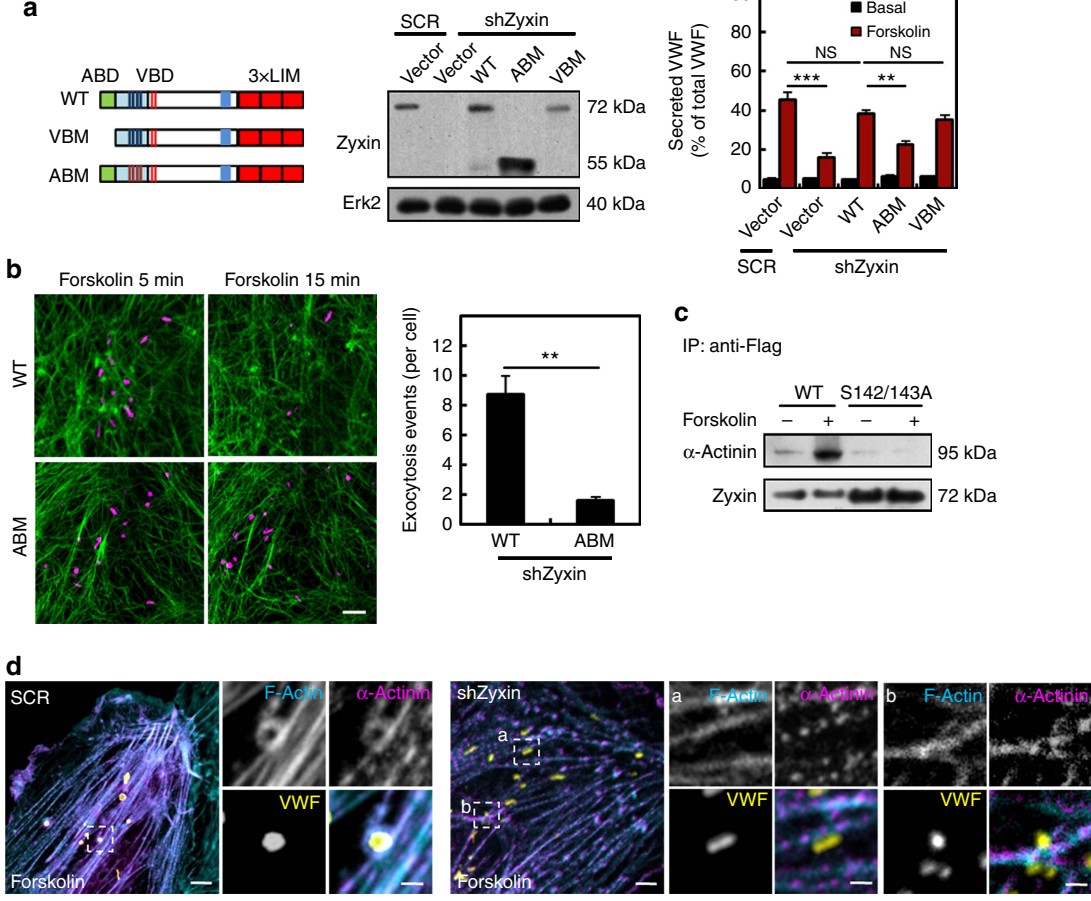

**Figure 5 | Zyxin-mediated VWF secretion depends on the interaction with α-actinin. (a)** Left: cartoons for the constructs of zyxin and its α-actinin-binding mutant (ABM) and its VASP-binding mutant (VBM). Middle: western blotting of zyxin and its mutants. Right: ELISA of VWF secreted from SCR and shZyxin-targeted HUVECs that were rescued by constructs as indicated with forskolin stimulation ($n = 14$, \*\*$P < 0.01$, \*\*\*$P < 0.001$, NS $P > 0.05$, 3 independent experiments). **(b)** Left: HUVECs co-expressing mCherry-PSL-lum (magenta) and GFP-Lifeact (green) were infected with shZyxin, followed by introduction of WT zyxin (WT) or the zyxin ABM. The still images show the first (5 min) and last (15 min) imaging time points. ($n = 4$, 2 independent experiments). Right: the bar graph shows the average number of exocytosis events in each WT or ABM rescued cell ($n = 4$, \*\*$P < 0.01$, 2 independent experiments). **(c)** Immunoprecipitation western blotting of α-actinin with the indicated antibodies, using cell lysates of HUVECs expressing WT zyxin or the zyxin S142/143A mutant with forskolin stimulation. **(d)** Confocal images of immunofluorescent staining of SCR and shZyxin-targeted HUVECs with forskolin stimulation. The cells were co-stained for actin (cyan), VWF (yellow) and α-actinin (magenta). The images on the right panels correspond to the magnified insets. All error bars represent s.d. Student's *t*-test was used for statistical analysis. Scale bars, 500 nm in magnified insets of (**d**); 2 μm in the widefield of (**b**,**d**). The uncropped scans of western blottings were shown in Supplementary Fig. 9.

application in patients with VWD and mild haemophilia A following administration of epinephrine or the vasopressin analog DDAVP[5]. Thus this study may provide an important therapeutic target for the treatment of related diseases.

In summary, we define FA protein zyxin as a new physiological regulator of endothelial VWF secretion and demonstrate a zyxin-mediated mechanism for the formation of actin frameworks on exocytic granules from the local cortical actin network.

## Methods

**Reagents.** Forskolin (F6886), epinephrine (E4642), phorbol-12-myristate-13-acetate (P1585), heparin (H3149), 4′, 6-diamidino-2 phenylindole dihydrochloride (DAPI, D8417), dimethylsulfoxide (D2650) and anti-flag M2 affinity gel (A2220) were from Sigma. Rabbit anti-VWF antibody (A0082) was from Dako. Rabbit polyclonal antibody to ERK2 (sc-292838) was from Santa Cruz Biotechnology. Rabbit and mouse monoclonal antibodies against zyxin (ab109316, ab50391), mouse monoclonal antibodies to vinculin (ab18058) and mouse monoclonal antibody to mouse α-actinin (ab18061) were from Abcam. Mouse monoclonal antibody to human P-selectin (Q01102) was from R&D Systems. Rabbit polyclonal antibody to phospho-zyxin Ser 142/143 (#4863) was from Cell Signaling

Technology. Alexa Fluor 488-nm-conjugated phalloidin (A12379) and Alexa Fluor-conjugated goat anti-IgG antibodies (A11008, A27039, A21424, A21052) were from Invitrogen.

**DNA constructs.** The zyxin constructs tagged with GFP or mCherry at the C-terminus were generated by amplifying gene-specific cDNAs and ligating them into pRetroQ vectors (with GFP/mCherry, Addgene). Halo-Zyxin was constructed by subcloning the Halo tag from pSCS3V21C-Halo and replacing the mCherry sequence in the mCherry-zyxin construct with the Halo sequence. The constructs of the WT and the S142/143A mutant of zyxin tagged with Flag at the N-terminus were subcloned from GFP-zyxin into pBabe-puro vectors. The zyxin S142/143A mutant was generated by overlap extension PCR. The zyxin α-actinin-binding mutant was generated by subcloning the 127–1,719 bp segment of zyxin and ligating it into pBabe-puro vectors. The zyxin VASP-binding mutant was generated by the synthesis of double-strand oligos of 174 bp, which replaced the 211–213, 277–279, 310–312 and 340–342 bp segments of zyxin to change phenylalanine to alanine, and ligation into pBabe-puro vectors (Addgene). The construct for a fusion protein of the luminal part of P-selectin with mCherry at the C-terminus (mCherry-PSL-lum) was generated by amplifying the luminal part of P-selectin[11] and ligating it into the pRetroQ vector (with mCherry). The GFP-VWF construct was subcloned from an EGFP-VWF expression vector, which was a kind gift from Dr J. Voorberg, in pRetroQ vectors. The C terminal-tagged GFP-Lifeact construct

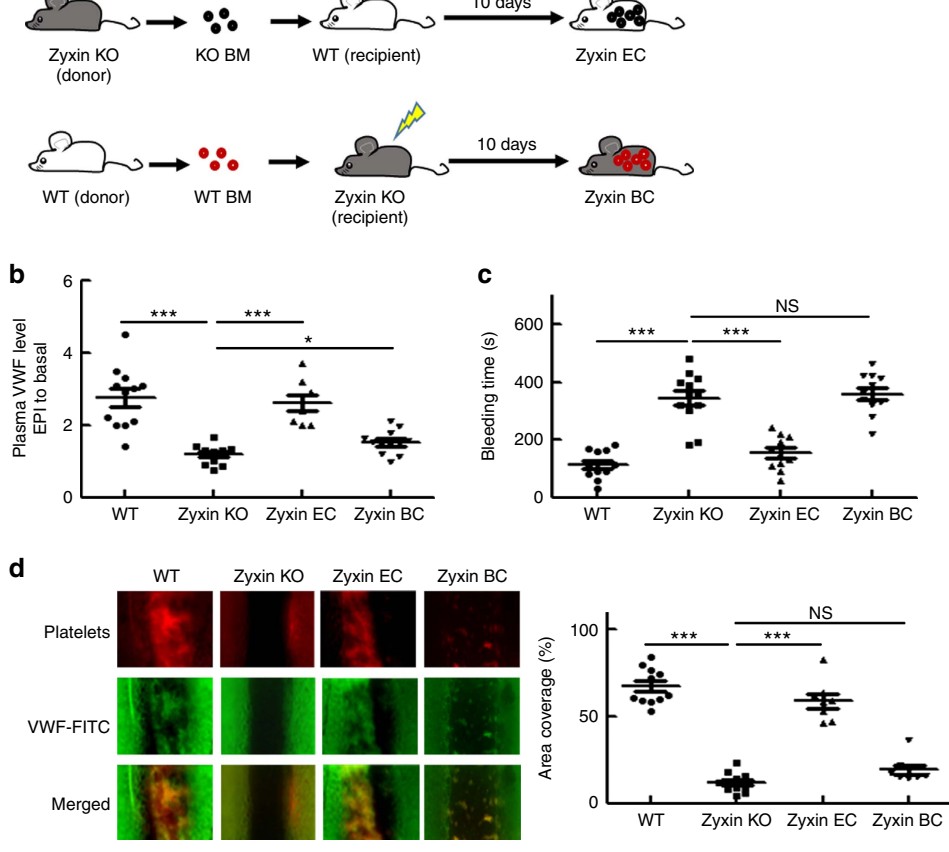

**Figure 6 | Zyxin-mediated endothelial exocytosis is required for vascular thrombosis and haemostasis in mice.** (**a**) The scheme of cross marrow transplantation for Zyxin BC and Zyxin EC mice. (**b**) Normalized level of plasma VWF in WT ($n = 12$), Zyxin KO ($n = 12$), Zyxin EC ($n = 12$) and ZyxinBC ($n = 8$) mice after epinephrine (EPI) stimulation (*$P < 0.05$, ***$P < 0.001$). (**c**) Blood bleeding time of WT ($n = 12$), Zyxin KO ($n = 12$), Zyxin EC ($n = 11$) and ZyxinBC ($n = 12$) mice after epinephrine stimulation (***$P < 0.001$, NS > 0.05). (**d**) Thrombus formation in the mesenteric vessels of WT ($n = 11$), Zyxin KO ($n = 11$), Zyxin EC ($n = 8$) and ZyxinBC ($n = 8$) mice. The thrombus is indicated with Rhodamine-labelled platelets and FITC-conjugated anti-VWF antibody. The graph showed the coverage of the area in the thrombus-formation assay. (***$P < 0.001$, NS > 0.05). All error bars represent s.e.m. Student's $t$-test was used for statistical analysis.

was generated by the synthesis of double-strand oligos of Lifeact and ligation into the pRetroQ vector.

A commercial lentiviral system from Sigma was used to silence 37 genes, including those for zyxin, a-actinin and VASP. The target and control scrambled sequences were selected from the human shRNA library of Sigma (http://www.sigmaaldrich.com/china-mainland/zh/life-science/functional-genomics-and-rnai/SHRNA/library-information.html). In addition to the commercial shRNA library, another two shRNA constructs of zyxin were generated by the synthesis of double-strand oligos and ligation into pLKO.1 vectors with the targeting sequences 5′-CTGGGTCACAACCAAATCA-3′ and 5′-GGCGACGAA TTGACCAAAGCA-3′.

**Cell culture.** HUVECs were isolated and cultured in M199 medium (Gibco, 31100035) supplemented with fibroblast growth factor, heparin and 20% fetal bovine serum (FBS) or ECM medium containing EC growth supplement and 10% FBS. Cells were tested for mycoplama contamination before use. Cells from passages 3–6 were serum-starved in 1% bovine serum albumin (BSA) for 4 h before stimulation for the VWF enzyme-linked immunosorbent assay (ELISA) test. 293 T cells (ATCC, CRL-3216) were cultured in DMEM containing 10% FBS.

**Virus preparation and infection.** Preparations of lenti- or retro-virus were made in 293 T cells. For transfection, 293 T cells were seeded at $1 \times 10^4$ cells cm$^{-2}$ in 2 ml complete medium and grown for 24 h until 80–90% confluent. Transfection complexes (polyethylenimine and target and packaging plasmids) were formed at room temperature in serum-free medium before drop-wise addition to the cells, followed by incubation for 5 h and replacement with complete medium for 48 h. The virus-containing supernatants were harvested. For infection, 3 ml of the virus-containing supernatant was mixed with 3 ml of fresh medium, and polybrene was added to a final concentration of 8 μg ml$^{-1}$. ECs were incubated in the

virus-containing medium for 48 h. The medium was replaced with culture medium after infection.

**ELISA analysis.** Confluent HUVECs were starved in serum-free M199 medium supplemented with 1% BSA for 4 h. For inhibitor treatment, the cells were incubated with PKA-specific inhibitor (2 μmol l$^{-1}$) or brefeldin A (EPAC pathway inhibitor, 5 μmol l$^{-1}$) for 15 min before determining VWF release. For stimulation, the cells were rinsed and incubated in serum-free medium in the presence or absence of forskolin (10 μmol l$^{-1}$), epinephrine (100 μmol l$^{-1}$) or other factors at the indicated concentrations for 1 h. The medium was collected and the remaining cells were lysed to determine total VWF levels. Relative amounts of VWF were determined by ELISA. Ninety-six-well plates were coated with the anti-VWF antibody (1:1,000) and blocked before addition of the basal, stimulated and lysate samples alongside serially diluted human plasma as a VWF standard. The plates were washed, and the horseradish peroxidase (HRP)-conjugated anti-VWF antibody (1:2,000) was added. After further rounds of washing, the plates were developed with o-phenylenediamine dihydrochloride and hydrogen peroxide in citrate phosphate buffer. Absorbance was analysed at 490 nm. Basal and stimulated release is presented as a percentage of the total VWF present in the cells.

**Western blotting and immunoprecipitation analysis.** Confluent HUVECs were starved in serum-free M199 medium supplemented with 1% BSA for 16 h. The cells were stimulated with forskolin (10 μmol l$^{-1}$) or epinephrine (100 μmol l$^{-1}$), washed twice in ice-cold PBS and lysed in buffer containing HEPES (50 mmol l$^{-1}$), NaCl (150 mmol l$^{-1}$), 1% Triton X-100, 10% glycerol, MgCl$_2$ (1.5 mmol l$^{-1}$), EGTA (1 mmol l$^{-1}$), EDTA (5 mmol l$^{-1}$), Na4P$_2$O$_7$ (0.27 g ml$^{-1}$), aprotinin (5 μg ml$^{-1}$), prostatin A (1 μg ml$^{-1}$), antipan (1 μg ml$^{-1}$), leupeptin (10 μg ml$^{-1}$), phenylmethyl sulfonyl fluoride (1 mg ml$^{-1}$), β-glycerol phosphate

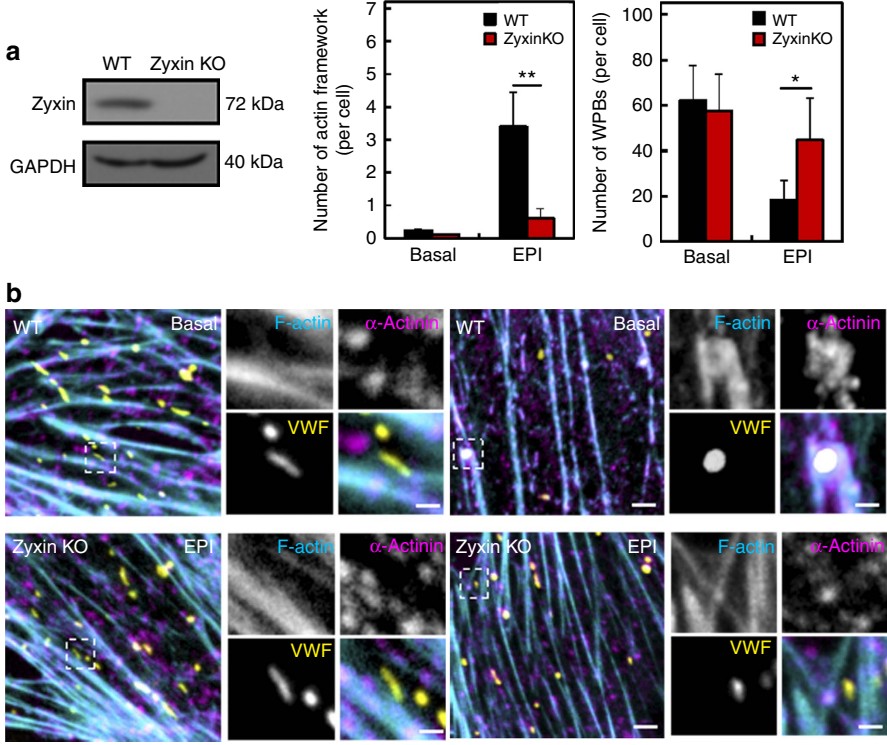

**Figure 7 | The actin framework formation and WPB exocytosis is impaired in zyxin-deficient ECs.** (**a**) Western blotting of zyxin in WT and zyxin KO mice. Protein sample were collected from the heart of WT and zyxin KO mice. (**b**) Confocal images of ECs isolated from the hearts of WT and zyxin KO mice before and after epinephrine stimulation (EPI). The ECs were immunostained with antibodies against VWF (yellow) or α-actinin (magenta). F-actin was labelled with Phalloidin (cyan). The images on the right panels correspond to the magnified insets. (**c**) Quantitative analysis of (**b**): Left, the number of actin framework-coupled WPBs per cell; Right: the total number of WPBs containing both framework-coupled and uncoupled WPBs per cell. ($n = 21$, \*\*$P < 0.01$, \*$P < 0.05$, 3 independent experiments). Scale bar, 500 nm in magnified insets of (**b**); 2 μm in the widefield of (**b**). All error bars represent s.d. The uncropped scans of western blottings were shown in Supplementary Fig. 9.

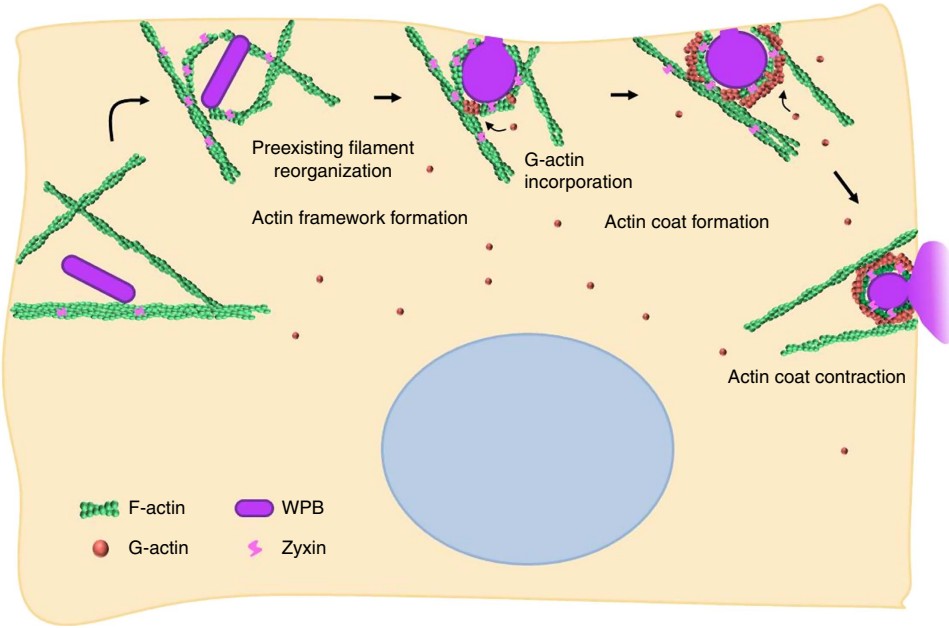

**Figure 8 | The model for exocytotic granules-coupled actin framework formation and subsequent actin coating.** Upon stimulation, the pre-existing filaments were reorganized to form the actin frameworks around exocytic granules, limiting the granules' movement and promoting them close to the plasma membrane. Subsequently, actin monomers are recruited from the cytosol and form coat structures around granules within the actin frameworks upon fusion. Cells may use such a synergetic strategy to promote exocytosis effectively and precisely.

(2 mmol l$^{-1}$), NaF (10 mmol l$^{-1}$) and Na$_3$VO$_4$ (2 mmol l$^{-1}$). The lysates were fractionated on 8% SDS–polyacrylamide gel electrophoresis, followed by standard western blotting analysis. For immunoprecipitation, the lysates were precipitated with antibodies at 4 °C for 7 h, incubated with 20 µl Protein A beads or anti-flag M2 affinity gel for an additional 2 h, denatured and then fractionated on 8% SDS–polyacrylamide gel electrophoresis for western blottings. The following antibodies were used for western blotting and immunoprecipitation: rabbit polyclonal antibody to ERK2 (Santa Cruz, sc-292838, 1:2,000), rabbit monoclonal antibodies against zyxin (Abcam, ab109316, 1:2,000), mouse monoclonal antibody to α-actinin (Abcam, ab18061, 1:1,000), rabbit polyclonal antibody to phospho-zyxin Ser 142/143 (CST, #4863, 1:1,000), and HRP-conjugated secondary antibodies (anti-rabbit NA9340, anti-mouse NA9310, GE, 1:5,000).

**Immunofluorescence staining and imaging.** Confluent ECs with or without forskolin stimulation (10 µmol l$^{-1}$) were washed twice with cold PBS, fixed and permeabilized with 3.7% formaldehyde and 0.1% Triton-X100 in cytoskeleton buffer (in mmol l$^{-1}$: 10 MES pH 6.1, 3 MgCl2, 138 KCl, and 2 EGTA) supplemented with 0.32 M sucrose for 10 min at room temperature and fixed for a second time with 3.7% formaldehyde in cytoskeleton buffer for 10 min at room temperature. For immunofluorescence labelling of VWF, vinculin, zyxin and α-actinin 1, fixed samples were permeabilized and blocked in blocking buffer (5% w/v BSA, 0.2% v/v saponin in PBS) for 1 h and then stained with primary antibodies (described below) in blocking buffer for 1 h at room temperature. The samples were washed twice and then stained with secondary antibodies (described below) in blocking buffer for 1 h at room temperature and washed twice. Finally, the cells were counterstained with DAPI. The images were acquired under a 40 × oil-immersion lens and analysed by laser scanning confocal microscopy (Zeiss LSM 510). Actin filaments were labelled with Alexa Fluor 488-conjugated phalloidin (A12379, 1:1,000) (0.5% v/v) along with secondary antibodies for 1 h at room temperature. The following primary antibodies were used for immuno-fluorescent staining: polyclonal antibody against VWF (Dako, A0082, 1:10,000), mouse monoclonal antibody to zyxin (Abcam, ab50391, 1:200), mouse monoclonal antibody to α-actinin (Abcam, ab18061, 1:100), and secondary antibodies from Thermo: goat anti-rabbit Alexa Fluor 555 (A27039, 1:1,000) and Alexa Fluor 647 (A21052, 1:1,000) labelled secondary antibodies.

**Spinning disk confocal microscope imaging.** HUVECs were infected with a retrovirus system to express different fluorescent fusion proteins. For spinning disk confocal imaging, the infected cells were plated on coverslips and imaged in a temperature-controlled chamber at 37 °C with forskolin (10 µmol l$^{-1}$) stimulation. Cells were visualized using a 100 × oil-immersion lens. Z-stacks were acquired at a step size of 500 nm using a piezo every 5 s for 15 min. Images were processed and analysed in ImageJ. The maximum intensity projection was generated using the Z-projection tool in ImageJ.

**Live cell imaging with high NA TIRF-SIM.** Infected HUVECs were plated on high-index coverslips 24 h before imaging. Cells were incubated with imaging buffer containing phenol-red-free M199 with 1% BSA. Zyxin-Halo was labelled by the JF549-HaloTag ligand (20 nmol l$^{-1}$)[26]. HUVECs were treated with the JF549-HaloTag ligand for 30 min and then washed twice with PBS before imaging. The images were acquired 5 min after stimulation with phorbol-12-myristate-13-acetate (160 nmol l$^{-1}$), forskolin (10 µmol l$^{-1}$) or epinephrine (100 µmol l$^{-1}$). All high NA TIRF-SIM images (512 × 512 pixels) were acquired with a 1.57-NA objective (Zeiss) under the physiological conditions of 37 °C and 5% CO$_2$. For multicolour imaging, each channel was captured on a separate sCMOS camera. Nine raw images were acquired at each excitation wavelength (488 and 561 nm) before moving to the next, and then this series was repeated every 1 or 3 s. Raw images were reconstructed as previously described[21] and linear adjustments and gamma were made using ImageJ.

**Fluorescent G-actin-labelling assay.** ECs were plated on collagen-coated coverslips and then were imaged for 1 min at 5 s per frame speed. Then cells were permeabilized and labelled by perfusing the saponin/G-actin solution containing forskolin/IBMX at the same time. After 5 min, the images were continually acquired for 10 min. The labelling and permeabilizing mixture contains 0.01% saponin and 0.5 µmol l$^{-1}$ AF647-actin in the buffer as described before[19,21], containing 10 µmol l$^{-1}$ forskolin and 0.1 mmol l$^{-1}$ IBMX. The intensity of the incorporated G-actin was statistically analysed using ImageJ by measuring the its intensity on the actin coat. The fluorescent intensity was normalized by the intensity of the first time point.

**Analysis of movies.** ImageJ was used to determine the mean fluorescence intensity of the infected fluorescent proteins during exocytosis. The normalized fluorescent intensity was defined as the ratio of the intensity at each time point to the first time point. The fluorescent intensity of actin was defined as that of the 1 µm$^2$ region around a WPB. Alignment was conducted according to bead imaging with pixel accuracy using ImageJ.

**Cross bone marrow transplantation.** Zyxin KO mice were from C57BL/6J background (from Jackson Laboratory) and were interbred with WT C57BL/6J mice (from Vital River Ltd. Co.) to generate zyxin KO and WT control littermate mice. One week before irradiation, the recipient mice on the C57BL6/J background were given acidified, antibiotic water. Eight-to-10-week-old mice of both genders (recipients) were conditioned with a lethal dose of 1000 rads total body irradiation. This is given as two irradiation doses with a 3-h interval. Then 12 h later, the bone marrow cells were flushed from the femurs of the donors and resuspended in serum-free RPMI. The appropriate number of bone marrow cells were transplanted to the recipients through intrabone injection. The experiments were carried out about 10 days after transplantation[34]. The reconstitution efficiency was assessed by fluorescence-activated cell sorting with a Verse (BD Biosciences, San Jose, CA, USA) for detecting zyxin-expressing cells. The peripheral blood samples were harvested from each mouse and BCs were resuspened with RBC lysis buffer (Stemcell, Cat. #20120) to lyse the erythrocytes and then fixed in 1% paraformaldehyde for 10 min at room temperature, followed by permeabilization with 0.1% saponin (Sigma-Aldrich, St Louis, MO, USA) for 5 min at room temperature. For immunostaining intracellular zyxin, a rabbit monoclonal antibody (Abcam, Cat. #ab109316, 1:50) was used as the primary antibody and AF488-conjugated goat anti-rabbit IgG (Thermo, A11008, 1:1000) was used as the secondary antibody. For a negative control, a separate set of cells were stained with an isotype control antibody.

**Measurement of tail bleeding time.** Eight to ten-week-old WT and zyxin KO mice of both genders (C57BL/6J background) were anaesthetized by i.p. injection of pentobarbital sodium (0.15 ml per 10 g body weight), and the distal 3 mm segment of the tail was amputated with a scalpel. Immediately, the tail was placed in PBS buffer and maintained at 37 °C. Bleeding time was recorded from the time of injury until blood loss ceased. The assay was terminated manually after 10 min. Bleeding times > 10 min were recorded as 10 min[35]. Mice at 8–10 weeks of age were randomly grouped and used; grouped statistical analysis was performed using Graphpad.

**Measurement of plasma VWF.** Eight-to-ten-week-old mice of both genders (C57BL/6J background) were stimulated with 0.5 mg kg$^{-1}$ epinephrine i.p. Before and after 30 min of stimulation, peripheral blood samples were collected from tail vein into 50 µl of 3.2% sodium citrate. The plasma was obtained by collecting the supernatant after centrifugation at 200g for 10 min. Plasma VWF levels were measured using ELISA. A rabbit polyclonal antibody against VWF (Dako, A0082, 1:2,000) and a HRP-conjugated antibody against VWF (Dako, 1:3,000) were used for capture and detection. Pooled plasma was defined as a mixture of plasma from five WT C57BL/6J mice. The levels of plasma VWF are shown as the ratio to the pooled plasma[33]. The statistical analysis was performed using Graphpad. Animals were randomly assigned by independent persons, and the data acquisition and analysis were being blinded to the experimental groups.

**Thrombus formation in mesenteric vessels.** Peripheral blood was harvested from tail tip, centrifuged at 250g for 10 min and platelet-rich plasma was gently transferred to a fresh tube. Platelets were labelled with rhodamine (2.5 mg l$^{-1}$) and adjusted to a final concentration of 200 × 10$^6$ per 250 µl. Eight-to-10-week-old mice of both genders (C57BL/6J background) were anaesthetized with pentobarbital sodium (0.15 ml per 10 g body weight, i.p.). Antibody against VWF was conjugated with fluorescein isothiocyanate (FITC). Both rhodamine-labelled platelets and FITC-conjugated-anti-VWF (1 µg µl$^{-1}$) were injected into the tail vein. The mice were stimulated with epinephrine 15 min before imaging. The mesentery was externalized through a midline abdominal incision. Vessels 3 mm in diameter were visualized at 10 × magnification with an intravital microscope. Then a filter paper (2 × 1 mm) saturated with 20% FeCl$_3$ was applied to the vessels for 2 min. The vessels were monitored for 20 min after FeCl$_3$ application or until complete occlusion (blood flow stopped for 1 min). Images were acquired every 15 s. Three Z-stacks were acquired at every time point at steps of 60 µm. The images shown represent the maximum intensity projection of all Z-stacks. A thrombus was defined in terms of the fluorescence of aggregated platelets. All animal procedures were carried out according to the rules of Association for Assessment and Accreditation of Laboratory Animal Care International and were approved by the Institutional Animal Care and Use Committee of Peking University.

**Data availability.** All data supporting the findings are included in this published article (and its Supplementary files), and all relevant materials are available from the authors upon request.

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

## Acknowledgements

We thank J. Voorberg (Sanquin Research Laboratory, Amsterdam, The Netherlands) for kindly providing GFP-VWF. We thank C. Shan for excellent technical assistance. We are also grateful to H.P. Cheng, I.C. Bruce and P.Y. Xu for reading the manuscript and offering valuable comments. J. L. was supported by research grants from the National Science Funds (No. 91339111, No. 81470298 and Project 31521062), the Major State Basic Research Development Program of China (No. 2012CB945103) and National Science and Technology Support Project 2014BAI02B01. D.L. was funded by National Key Research and Development program (Grant No: 2016YFA0500203) of the Ministry of Science and Technology. D.L. and E.B. were funded by the Howard Hughes Medical Institute (HHMI).

## Author contributions

J.L., D.L. and E.B. designed the experiments and wrote the manuscript. X. Han and P.L. carried out the experiments, analysed data and wrote the manuscript. Y.S. and L.C. designed the experiments and edited the manuscript. Z.Y., Y.H., X.W., X. Huang, X.K., Q.D., A.H. and X. Hu performed the experiments.

## Additional information

**Competing financial interests:** The authors declare no competing financial interests.

