## [Peer Review File · Nature Communications]

Reviewers' comments:

Reviewer #1 (expert in exocytosis)

Remarks to the Author:

In this work by Han et al. the process of VWF secretion is studied in endothelial cells. The authors find through a shRNA screen that specific components of focal adhesions influence VWF secretion. High-resolution high NA TIRF-SIM imaging shows that actin forms around these vesicles prior to fusion. The authors further show that the removal of zyxin, mutation of zyxin, or the removal of alpha-actinin perturbs the formation of these actin coats and inhibits exocytosis. Exocytosis of VWF is less pronounced in mutant cells and in animals that have mutations to these genes. In general I find this work quite nice. It is a good combination of a genetic/protein screen, high resolution imaging, and an animal model where a known protein is shown to be critical to an important physiological process--- Namely, that the focal adhesion protein zyxin is involved in actin coat formation around exocytic vesicles in endothelial cells.

My primary concern with the paper is with one of the main interpretations of the TIRF-SIM data and the final model shown in Figure 7B. The authors make the argument here and in the discussion that their data does not support the classic model of "de novo" formation of actin filaments from a monomeric pool of actin but instead shows a re-arrangement of existing cortex filaments around the exocytic vesicle. In my opinion the authors show no evidence for this new alternative model. I see that in some of their movies and figures actin appears to start forming from an existing filament but this does not imply that new actin is not recruited from the cytoplasm to build around this existing filament seed. Without some proof of this model I don't think the authors should emphasize this point and this model. They should be more flexible (or open) in their interpretation of the imaging data. It is possible that a mixed model exists for example.

A FRAP (or photo-activation) experiment (or something similar) would be needed to experimentally prove their final model 7B. Also, the noise present in the TIRF-SIM data (very obvious pattern noise and the occasional reconstruction errors (eg. waffle patterns in movies) makes it very difficult to interpret any of the local or global intensity measurements over time in a quantitative way.

Otherwise I think the paper is nicely done with an interesting finding that is useful for this field.

Reviewer #2 (expert in exocytosis)

Remarks to the Author:

Review of "Zyxin regulates endothelial von Willebrand factor secretion by mediating fusion independent assembly of actin-coats on granules" by Han et al.

The process of exocytosis-linked actin "coating" was originally described more than a decade ago in studies of cortical granule exocytosis in frog eggs (Nat Cell Biol. 2003. 5(8):727-32.). The general idea is that following fusion of certain secretory granules-particularly those that contain large, insoluble components (Mol Biol Cell. 2006. 17(4):1495-502)-is rapidly followed by assembly of actin filaments (F-actin) on the cytoplasmic face of the secretory granule. The coating was proposed to promote internalization of granule membrane after exocytosis (Nat Cell Biol. 2003. 5(8):727-32; Mol Biol Cell. 2006. 17(4):1495-502) and exocytosis itself, by virtue of compressing the exocytotic compartment (Mol Biol Cell. 2006 Apr;17(4):1495-502.) with the latter notion being confirmed by elegant studies in several systems (e.g. J Cell Biol. 2011. 194(4):613-29; J Cell Sci. 2012 125(Pt 11):2765-74).

In the current study, Han et al reveal a role of the actin binding protein, zyxin (best known as a component of focal adhesions in exocytosis of von Willebrand factor). The authors further report that actin coating is not, as previously reported, dependent on either de novo actin assembly or fusion of the secretory granule with the plasma membrane but instead occurs by reorganization of preexisting cortical F-actin.

The ideas presented here are provocative, but based on the data presented, an alternative explanation of the results seems equally, if not more plausible: first, that zyxin is involved in positioning the WBP in close proximity to the plasma membrane rather than coat formation; second, that this positioning is accompanied by reorganization of cortical F-actin as the WBP moves into the cortical F-actin network; third, that the deficits in VWF reflect failed fusion of the WBPs with the plasma membrane rather than failed coat formation per se.

Why this interpretation rather than that proposed by the authors? Essentially it comes down to the images provided by the authors, the experiments performed (or not performed by the authors) and what coating is supposed to represent. These points are enlarged upon below.

If the role of zyxin is to promote coat formation or reorganization, the predicted result of loss of zyxin is not having more WBPs after stimulation (as found here) but WBPs that less efficiently release their contents. That is, coats are not supposed to be involved in granule-PM fusion, but in promoting efficient release of granule contents after fusion. In contrast, if the role is to promote positioning of the WBPs rather than coat formation or function per se, the result is exactly what is seen here since failure of positioning should leave more intact WBPs.

The images shown by the authors are higher resolution than anything else published in the small but growing body of coating literature. However, they are still well above the limit of resolution needed to visualize individual actin filaments (4 nm diam) and they cannot by themselves distinguish de novo assembly versus reorganization of preexisting filaments. For example, in Figure 2A the authors show an actin filament that extends to the right over time and use this as evidence that the coats are assembling from preexisting F-actin rather than de novo assembly. However, there is no way to determine from this movie/montage whether the extension reflects polymerization (ie de novo assembly) or translocation (ie reorganization). In contrast, in the studies where it was originally reported that actin coats form at least in part from de novo assembly (Nat Cell Biol. 2003. 5(8):727-32; this paper should probably be cited, particularly if the authors are going to strongly argue against de novo assembly), two actin probes were employed-fluorescent phalloidin, which binds f-actin but not g-actin, and fluorescent g-actin. Actin coats incorporated fluorescent g-actin before the phalloidin, a result indicative of de novo assembly. Additionally, coat assembly was shown to be dependent on N-WASP, a factor that promotes de novo actin assembly.

There is an additional problem, namely, whether what the authors are calling coats really reflect the term as generally employed. By definition, an F-actin coat surrounds (coats) the cytoplasmic face of the secretory granule. To my eye, the montages and the movies of the authors seem to show that before exocytosis, there are actin filaments or cables associated with parts of the cytoplasmic face of the WBPs before exocytosis, but there is nothing that looks like a coat-ie a complete enclosure of the exocytotic compartment. Now, you might argue that the previous notion of a coat stems simply from lower resolution of the older such that a structure that seems to be coating the granule surface is actually only a few filaments or cables touching here and there, as seen in this study. However, after exocytosis, structures that look far more like the coats previously reported are seen in this study. Thus, an apparently complete coating of the cytoplasmic face of the WPB is apparent, even using super resolution, but only after exocytosis (see also below). It therefore follows that the actin associated with the WBPs before exocytosis is not a coat since, if it were, it would be elongate, like the

WPBs. Bottom line: If the authors really wish to test the relative roles of reorganization of cortical F-actin versus de novo assembly they will have to use an approach that lets them distinguish between preexisting actin filaments versus newly assembled actin filaments. This could be done as previously or it could be done using speckle microscopy or some other means (eg photactivation) of distinguishing between assembling and moving F-actin.

The second major claim made by the authors, that coating does not depend on fusion, also goes untested, except insofar as the authors state that since they see what they interpret as the onset of coating before fusion, fusion must not be required. This is not a good argument, based on the observations presented above. In the original studies where it was demonstrated that coating was critically linked to secretory granule fusion with the plasma membrane, it was shown that a) coating is limited to those granules that fuse with the plasma membrane (Mol Biol Cell. 2006. 17(4):1495-502) and b) that inhibition of secretory granule fusion with the plasma membrane via SNARE inactivation blocks coating in response to secretory stimuli such as calcium elevation (Nat Cell Biol. 2007 9(2):149-59). If the authors really wish to test the importance of fusion, they would need to perform experiments in which they disable the SNAREs and then treat the cells with a secretagogue and monitor coat assembly.

This is not to suggest that the findings are not novel. Plenty of people have alluded to changes in cortical F-actin that may accompany close apposition of secretory vesicles to the plasma membrane but the results here provide the first dynamic close ups of this process. In addition, a potential role of zyxin in exocytosis is important, regardless of how it actually works. But the authors claims go far beyond these basic findings into areas that their data really do not address.

Reviewer #3 (expert in thrombosis)

Remarks to the Author:

In this article by Hun X, et al on zyxin biology in Weibel-Palade body (WPB) exocytosis, the authors focused on a new technology, total internal reflection fluorescence structured illumination microscopy (TIRF-SIM) to visualize the dynamics of cortical actin filaments and the behavior of exocytosing granules-WPB, and found that WPB exocytosis process is dependent on zyxin and an interaction with α -actinin. Their findings are novel to indicate that zyxin is a regulator of WPB secretion and hemostasis and thrombosis and provided important new insights into the late events in how exocytosis occurs.

WPB were first noted in endothelial cells by electron microscopy almost 50 years ago by Weibel and Palade. They are important stores of high molecular weight von Willebrand factor (VWF) and important in hemostasis and in several diseases like thrombotic thrombocytopenic purpura. As noted by the authors, induced exocytosis of WPBs has been used to care for many patients with von Willebrand Disease and mild hemophilia A.

Major comments:

The physical characterization studies of the events in exocytosis of the WPB are the strength of the manuscript. The authors need to discuss how widespread the exocytotic events seen with the WPB apply to other granular pools in the body even briefly such as on coat color and neurodevelopment. For a hemostatic aspect, does the zyxin pathway have a role in alpha-granule release from platelets and the degranulation of monocytes and leucocytes because both sets of pathways would influence interpretation of the hemostasis and thrombosis outcomes seen and may also influence the levels of von Willebrand factor circulating in the plasma of the zyxin knockout mouse. Two approaches can be followed to answer this question of contribution by the endothelial defect to the overall hemostatic/thrombotic defect. Either an endothelial-specific knockout or cross bone marrow

transplantation studies of the zyxin knockout and wildtype controls would answer the question.

Minor comments:

1. Abstract and throughout the paper. Please define once what "end-stage" exocytosis refers to.
2. There are multiple zyxin shRNAs mentioned in this paper. For example, SHZyxin-1, and -2 in Figure 1. Then the authors use SHZyxin from Figure 2 and on. So which one is this SHZyxin? And why?
 2. Authors claim that Zyxin impairs WPB secretion; but in the experiment of Zyxin knockdown by shRNA, they indicated that shRNA decreased the number of WPB (Supplement Fig2). Does this mean Zyxin regulate WPB formation as well as secretion? Can the authors explain it?
3. Authors claim that zyxin co-localize with actin coats around WPB, however in Figure 3D. I cannot appreciate the co-localization in this particular image. Does it because the fixation affects the imaging? If not, please provide a more representative image to support the statement. Is there a program that can analyze this association in a non-biased fashion?
4. Figure 5D, authors chose a rod-shaped WPB in SHZyxin-HUVEC cells to show the absent of F-actin coat/actinin. This is not comparative to the exocytosing (round-shaped) WPB in SCR-HUVEC cells. Please provide the right evidence to support the statement.
5. Figure 6 has some inconsistent numbers for the tail vein studies.
6. There are other minor points that need to be addressed and an edited version of the manuscript is being returned (see attachment)

Response to the reviewers' comments

We thank the reviewers for carefully reading the manuscript and providing constructive criticism. Our point by point response is outlined below.

Response to Reviewer #1

General comments:

In this work by Han et al. the process of VWF secretion is studied in endothelial cells. The authors find through a shRNA screen that specific components of focal adhesions influence VWF secretion. High-resolution high NA TIRF-SIM imaging shows that actin forms around these vesicles prior to fusion. The authors further show that the removal of zyxin, mutation of zyxin, or the removal of alpha-actinin perturbs the formation of these actin coats and inhibits exocytosis. Exocytosis of VWF is less pronounced in mutant cells and in animals that have mutations to these genes. In general I find this work quite nice. It is a good combination of a genetic/protein screen, high resolution imaging, and an animal model where a known protein is shown to be critical to an important physiological process---Namely, that the focal adhesion protein zyxin is involved in actin coat formation around exocytic vesicles in endothelial cells.

We thank the reviewer for their enthusiasm and positive comments of our work.

Specific comments:

My primary concern with the paper is with one of the main interpretations of the TIRF-SIM data and the final model shown in Figure 7B. The authors make the argument here and in the discussion that their data does not support the classic model of "de novo" formation of actin filaments from a monomeric pool of actin but instead shows a re-arrangement of existing cortex filaments around the exocytic vesicle. In my opinion the authors show no evidence for this new alternative model. I see that in some of their movies and figures actin appears to start forming from an existing filament but this does not imply that new actin is not recruited from the cytoplasm to build around this existing filament seed. Without some proof of this model I don't think the authors should emphasize this point and this model. They should be more flexible (or open) in their interpretation of the imaging data. It is possible that a mixed model exists for example.

We are very grateful for this insightful comment. Indeed, the mixed model is supported by our new data (please see the response to the following comment). However, we admit that this evidence may be still not strong enough to support the proposed new model. Therefore, we deleted the model in Figure 7 and revised the

discussion accordingly. (Page 11)

A FRAP (or photo-activation) experiment (or something similar) would be needed to experimentally prove their final model 7B.

Following the reviewer's suggestion, we performed a similar experiment to examine whether new actin is recruited from the cytoplasm to sites of the reorganized actin framework, which may serve as "seeds" for subsequent actin coating around the exocytic vesicle. In this experiment, the G-actin was conjugated with Alex Fluor-647 to indicate the newly formed actin filaments. As shown in Supplemental Figure 5, G-actin was recruited to form a ring structure (coating) on WPBs upon fusion within the reorganized actin frameworks. The results support the comments of the reviewer that there is a mixed mechanism in the process. We described this data in the text (Page 6, line 15) and an appropriate discussion was added. (Page 11)

The noise present in the TIRF-SIM data (very obvious pattern noise and the occasional reconstruction errors (eg. waffle patterns in movies) makes it very difficult to interpret any of the local or global intensity measurements over time in a quantitative way.

We thank the reviewer for raising this issue. As we pointed out in our recent correspondence (Li D and Betzig E, Science, V.352, 527-b, 2016), the pattern noise in TIRF-SIM image is always associated with out-of-focus background, usually in the high refractive index sample region, leading to periodic patterns in reconstructed images. Since these artificial patterns are locally concentrated, we chose the region, which was not influenced by the out-of-focus background, to perform the quantification analysis.

As for the "waffle patterns" in movie, they were caused by the lower signal noise ratio due to the long duration of imaging, especially in the late time course of imaging. Nevertheless, we did not select these late time points to perform any quantification analysis, although we kept them in the movie to allow us to visualize what happened after fusion occurred.

Response to Reviewer #2

General comments:

Review of "Zyxin regulates endothelial von Willebrand factor secretion by mediating fusion independent assembly of actin-coats on granules" by Han et al. The process of exocytosis-linked actin "coating" was originally described more than a decade ago in studies of cortical granule exocytosis in frog eggs (Nat Cell Biol. 2003. 5(8):727-32.). The general idea is that following fusion of certain

secretory granules-particularly those that contain large, insoluble components (Mol Biol Cell. 2006. 17(4):1495-502)-is rapidly followed by assembly of actin filaments (F-actin) on the cytoplasmic face of the secretory granule. The coating was proposed to promote internalization of granule membrane after exocytosis (Nat Cell Biol. 2003. 5(8):727-32; Mol Biol Cell. 2006. 17(4):1495-502) and exocytosis itself, by virtue of compressing the exocytotic compartment (Mol Biol Cell. 2006 Apr;17(4):1495-502.) with the latter notion being confirmed by elegant studies in several systems (e.g. J Cell Biol. 2011. 194(4):613-29; J Cell Sci. 2012 125(Pt 11):2765-74). In the current study, Han et al reveal a role of the actin binding protein, zyxin (best known as a component of focal adhesions in exocytosis of von Willebrand factor. The authors further report that actin coating is not, as previously reported, dependent on either de novo actin assembly or fusion of the secretory granule with the plasma membrane but instead occurs by reorganization of preexisting cortical F-actin.

Specific comments:

The ideas presented here are provocative, but based on the data presented, an alternative explanation of the results seems equally, if not more plausible: first, that zyxin is involved in positioning the WBP in close proximity to the plasma membrane rather than coat formation; second, that this positioning is accompanied by reorganization of cortical F-actin as the WBP moves into the cortical F-actin network; third, that the deficits in VWF reflect failed fusion of the WBPs with the plasma membrane rather than failed coat formation per se. Why this interpretation rather than that proposed by the authors? Essentially it comes down to the images provided by the authors, the experiments performed (or not performed by the authors) and what coating is supposed to represent. These points are enlarged upon below. If the role of zyxin is to promote coat formation or reorganization, the predicted result of loss of zyxin is not having more WBPs after stimulation (as found here) but WBPs that less efficiently release their contents. That is, coats are not supposed to be involved in granule-PM fusion, but in promoting efficient release of granule contents after fusion. In contrast, if the role is to promote positioning of the WBPs rather than coat formation or function per se, the result is exactly what is seen here since failure of positioning should leave more intact WBPs.

We appreciate the reviewer's careful review and detailed comments. We agree with the reviewer that the reorganizing process of pre-existing actin filaments around the exocytic vesicle described in our paper is different from the "actin coating" originally described in studies of cortical granule exocytosis in frog eggs (Sokac AM, et al, Nat Cell Biol, 5: 727-732, 2003.). To avoid the confusion, we renamed the reorganizing structure of pre-existing actin filaments around the exocytic vesicle as "actin framework". In addition, we also agree with the reviewer that zyxin might be critical

for positioning the WBP in close proximity to the plasma membrane. Together with our new data (please see the response to the following comments and Supplemental Figure 9), we revised the discussion on Zyxin-mediated formation of actin framework in regulation of exocytosis. (Page 11)

The images shown by the authors are higher resolution than anything else published in the small but growing body of coating literature. However, they are still well above the limit of resolution needed to visualize individual actin filaments (4 nm diam) and they cannot by themselves distinguish de novo assembly versus reorganization of preexisting filaments. For example, in Figure 2A the authors show an actin filament that extends to the right over time and use this as evidence that the coats are assembling from preexisting F-actin rather than de novo assembly. However, there is no way to determine from this movie/montage whether the extension reflects polymerization (ie de novo assembly) or translocation (ie reorganization). In contrast, in the studies where it was originally reported that actin coats form at least in part from de novo assembly (Nat Cell Biol. 2003. 5(8):727-32; this paper should probably be cited, particularly if the authors are going to strongly argue against de novo assembly), two actin probes were employed-fluorescent phalloidin, which binds f-actin but not g-actin, and fluorescent g-actin. Actin coats incorporated fluorescent g-actin before the phalloidin, a result indicative of de novo assembly. Additionally, coat assembly was shown to be dependent on N-WASP, a factor that promotes de novo actin assembly.

Following the reviewer's suggestion, we cited the original report that actin coats form from *de novo* assembly (Sokac AM, *et al.*, Nat Cell Biol, 5: 727-732, 2003.). In addition, we performed a similar experiment to examine whether G-actin is recruited from the cytoplasm to sites of the reorganized actin framework around the exocytic vesicle. In this experiment, the G-actin was conjugated with Alex Fluor-647 to indicate the newly formed actin filaments. As shown in Supplemental Figure 9, G-actin was recruited to form a ring structure (coating) on WPBs upon fusion within the reorganized actin frameworks. We described this data in the text (see Page 6, line 15) and an appropriate discussion has been added. (Page 11)

The second major claim made by the authors, that coating does not depend on fusion, also goes untested, except insofar as the authors state that since they see what they interpret as the onset of coating before fusion, fusion must not be required. This is not a good argument, based on the observations presented above. In the original studies where it was demonstrated that coating was critically linked to secretory granule fusion with the plasma membrane, it was shown that a) coating is limited to those granules that fuse with the plasma membrane (Mol Biol Cell. 2006. 17(4):1495-502) and b) that inhibition of

secretory granule fusion with the plasma membrane via SNARE inactivation blocks coating in response to secretory stimuli such as calcium elevation (Nat Cell Biol. 2007 9(2):149-59). If the authors really wish to test the importance of fusion, they would need to perform experiments in which they disable the SNAREs and then treat the cells with a secretagogue and monitor coat assembly.

Since fusion is not the focus of this work, we didn't perform the experiments to study the importance of fusion in exocytosis-associated actin remodeling. In addition, we also revised the paper title accordingly.

This is not to suggest that the findings are not novel. Plenty of people have alluded to changes in cortical F-actin that may accompany close apposition of secretory vesicles to the plasma membrane but the results here provide the first dynamic close ups of this process. In addition, a potential role of zyxin in exocytosis is important, regardless of how it actually works. But the authors claims go far beyond these basic findings into areas that their data really do not address.

We appreciate the view of the reviewer with regard to the novelty and significance of our findings. To avoid the overstatement, we deleted the model in original Figure 7 and revised the discussion accordingly. (Page 11)

Response to Reviewer #3

General Comments:

In this article by Hun X, et al on zyxin biology in Weibel-Palade body (WPB) exocytosis, the authors focused on a new technology, total internal reflection fluorescence structured illumination microscopy (TIRF-SIM) to visualize the dynamics of cortical actin filaments and the behavior of exocytosing granules-WPB, and found that WPB exocytosis process is dependent on zyxin and an interaction with α -actinin. Their findings are novel to indicate that zyxin is a regulator of WPB secretion and hemostasis and thrombosis and provided important new insights into the late events in how exocytosis occurs. WPB were first noted in endothelial cells by electron microscopy almost 50 years ago by Weibel and Palade. They are important stores of high molecular weight von Willebrand factor (VWF) and important in hemostasis and in several diseases like thrombotic thrombocytopenic purpura. As noted by the authors, induced exocytosis of WPBs has been used to care for many patients with von Willebrand Disease and mild hemophilia A.

We thank the reviewer for the comments that this work is novel and has clinical relevance.

Major comments:

1. The physical characterization studies of the events in exocytosis of the WPB are the strength of the manuscript. The authors need to discuss how widespread the exocytotic events seen with the WPB apply to other granular pools in the body even briefly such as on coat color and neurodevelopment.

We thank the reviewer for this kind suggestion. We discussed this briefly in Page 11, Line 17.

“It is also worth noting that in addition to WPBs, secretory vesicles in other cell types, such as neurosecretory cells and melanocytes, need to undergo translocation from the cytosol to the plasma membrane, which was facilitated by the remodeling of cortical actin network.”

2. For a hemostatic aspect, does the zyxin pathway have a role in alpha-granule release from platelets and the degranulation of monocytes and leucocytes because both sets of pathways would influence interpretation of the hemostasis and thrombosis outcomes seen and may also influence the levels of von Willebrand factor circulating in the plasma of the zyxin knockout mouse. Either an endothelial-specific knockout or cross bone marrow transplantation studies of the zyxin knockout and wildtype controls would answer the question.

We really appreciate the valuable comments from the reviewer, which significantly deepened our observations. We chose cross bone marrow transplantation studies of the zyxin knockout and wildtype controls to address this question. As indicated in Figure 6, we established the murine models by cross bone marrow transplantation. Using these four murine models, we tested the plasma VWF level, bleeding time and the thrombus formation under epinephrine stimulation, and found that endothelial zyxin plays a critical role in maintaining the hemostasis under stress. We described the results in text (Page 9, Line 14), and added new Figure 7 to present the data on the role of zyxin in regulation of VWF secretion from isolated mouse ECs.

Minor comments:

1. Abstract and throughout the paper. Please define once what "end-stage" exocytosis refers to.

Thanks for the careful review. According to the literature, we would like to change the word “end-stage” into “final stage” because it has been more frequently used. The definition of the final stage of the regulated exocytosis is referred to as the docking,

the priming of the mature granules and the subsequent partial or total release of the secretory vesicle content. (Bond LM.*et al.*, *Biochem Soc Trans.* 39: 1115-1119, 2011). We briefly added this definition in the abstract where the final stage was first mentioned. (Page 2, Line 3)

2. There are multiple zyxin shRNAs mentioned in this paper. For example, SHZyxin-1, and -2 in Figure 1. Then the authors use SHZyxin from Figure 2 and on. So which one is this SHZyxin? And why?

We apologize for the confusion. As shown in Figure 1, the two SHRNAs of zyxin gene (SHZyxin-1, and -2) can both specifically decrease the expression level of zyxin. Thus, the SHZyxin-1 was used for the subsequent experiments. SHZyxin is the abbreviation of SHZyxin-1. We have indicated this fact when SHZyxin is used. (Page 4, Line 14)

3. Authors claim that Zyxin impairs WPB secretion; but in the experiment of Zyxin knockdown by shRNA, they indicated that shRNA decreased the number of WPB (Supplement Fig2). Does this mean Zyxin regulate WPB formation as well as secretion? Can the authors explain it?

As shown in the Supplement Figure 2, with respect to the unstimulated groups, the basal (without stimulation) number of WPBs between SCR and SHZyxin-targeted HUVECs exhibits no significant difference (62 ± 5 versus 57 ± 6 , $n=16$), indicating that the formation (at least the number) of WPB was not impaired by SHRNA. To further validate the conclusion that the deficiency of zyxin does not regulate the WPB formation, we harvest the total cell lysate of the SCR and SHZyxin cells in quiescent condition and then use ELISA to test the expression of VWF. As shown in the following graph, there is no significant difference in the expression levels of VWF between SCR and SHZyxin cells.

4. Authors claim that zyxin co-localize with actin coats around WPB, however in Figure 3D. I cannot appreciate the co-localization in this particular image. Does it because the fixation affects the imaging? If not, please provide a more representative image to support the statement. Is there a program that can analyze this association in a non-biased fashion?

We apologize for the bad quality of the image. The fixation may slightly affect the imaging. We optimized the staining protocol and got better imaging quality. So we changed the image in Figure 3D. We agree that a quantitative analysis of colocalization is beneficial, and thus used the intensity correlation analysis program in ImageJ to generate the analysis and the data is shown in revised Figure 3E.

5. Figure 5D, authors chose a rod-shaped WPB in SHZyxin-HUVEC cells to show the absent of F-actin coat/actinin. This is not comparative to the exocytosing (round-shaped) WPB in SCR-HUVEC cells. Please provide the right evidence to support the statement.

In zyxin deficient cells, most WPBs could not fuse with the plasma membrane and remain rod-shaped. The exocytosing WPBs was significantly less than the intact WPBs. As the rod-shaped WPB is more representative, we chose it to show the absence of F-actin coat/actinin. Following the reviewer's suggestion, the data on the comparison between exocytosing (round-shaped) WPBs in SCR-HUVECs and in SHZyxin-HUVECs was added in Figure 5D.

6. Figure 6 has some inconsistent numbers for the tail vein studies.

We apologize for this mistake. We corrected the numbers of mice regarding the murine model studies in the legend of Figure 6. We only did one tail clip per mouse and used different mice for each independent experiment.

7. There are other minor points that need to be addressed and an edited version of the manuscript is being returned (see attachment)

We thank the reviewer for the careful and kind comments. We revised all the grammatical mistakes according to the attachment.

As for the pattern of VWF polymer in Zyxin KO mice, which was questioned by the reviewer in the edited version of the manuscript, we performed the VWF polymer assay to test the quality of VWF. As shown in the following representative figure, there was no significant difference in the pattern of VWF multimer between WT and Zyxin KO even under stimulation by epinephrine (since the level of plasma VWF in Zyxin KO mice was significantly lower than that of WT mice, we adjusted the

amount of sample to the same level according to the ELISA results). Therefore, the deficiency of zyxin does not influence the pattern of VWF polymer.

REVIEWERS' COMMENTS:

Reviewer #1 (Remarks to the Author):

The authors have mostly answered my questions. I have a few additional comments.

I think the finding that Zyxin is involved in fusion of WPBs is novel and interesting and well supported. The model of actin recruitment the author's propose, however, is still not completely supported by their data. In fact, their new added data shown in Supplementary Figure 5 now supports the more traditional model that new actin monomers are recruited from the cytosol and "coat" large fusing vesicles during exocytosis. This coating model is standard in the field and supported by many other papers involving lamellar body exocytosis in alveolar cells as well cortical granules in eggs and other exocytic systems.

The new idea the authors present here is that the coating of WPBs occurs with the help of "pre-existing" actin filaments (abstract, page 4 line 2, page 6 line 3, page 11 line 2) that rearrange to (loosly???) surround vesicles before fusion. The authors try to distinguish what they are seeing from the standard model by renaming the actin structures an "actin framework".

I would recommend that the authors take a more inclusive tone in their discussion and clearly state that the actin recruitment and coating they observe with high-resolution TIRF-SIM imaging is likely a combination of new recruitment of actin monomers from the cytosol and a reorganization of existing filaments which fits with existing models. There is no strong data to support that the process they observe is different from what has been observed and presented in the past. It might even help to move Supplemental Figure 5 to the main text and figures as this is a key experiment.

These issues could be fixed by a careful re-working of their text with clear interpretations of their data and a speculative cartoon model.

Furthermore, the idea that the actin network is "strengthened", pg. 6. Line 9 or "structurally loose" pg 7. Line 10 are mechanical arguments. The authors provide no direct experimental evidence for this in this paper. Please remove these arguments or discuss the signals as "denser---i.e. more fluorescent signal etc". I mention this because something that is dispersed but rigid could be very strong and something dense and not networked could be very weak.

Reviewer #2 (Remarks to the Author):

The authors have addressed the experimental concerns I raised. It should be noted that the results have significantly changed the interpretation of the results in that, in contrast to the assertions made in the original submission, it now appears that actin coating occurs via de novo assembly and that the reorganization of the cortical f-actin detected by TIRF-SIM does not reflect coating per se. This does not bother me too much as the reorganization itself is novel and it is possible that the preexisting cortical actin contributes indirectly to coat formation.

Reviewer #3 (Remarks to the Author):

This manuscript on the role of zyxin in von Willebrand factor (vWF) extrusion from endothelial cells highly novel using state-of-the-art imaging technology to understand better the process of exocytosis is novel and advances the field of regulated extrusion of granules. The imaging studies as far as I can

interpret them support the conclusions. Also, the mice studies and biology studies of the released vWF are properly done and interpreted.

Major Concerns:

1. There is still not a full quantitation of data presented as images and videos. For example, In supplementary Fig 1, there is the claim that the Weibel Palade Bodies (WPB) lie along stress fibers by the number of cells examined and quantitation of that statement is missing. Similarly, other figures throughout the paper have biological statements made without a full quantitation. This is less an issue than the first submission, but still remains. Some of the now included quantitative data (e.g. Supplementary Figs 5B and 6B) are incompletely described methodologically or how the analysis was done.
2. Also lots of minor issues like inconsistent abbreviations and duplicated sentences are present. The supplemental videos need legends in the supplement section to explain what is being seen.

Minor Concerns:

1. While the manuscript focuses on WPB exocytosis, there are studies of platelet vWF and there is the low hanging fruit of whether vWF exocytosis in platelets is also zyxin dependent. People interested in vWF biology, separate from those interested in exocytosis would be interested in whether alpha granule release might also follow the same pathway and release of vWF following platelet activation might add to this manuscript.
2. I will return an annotated version of the manuscript with many minor/moderate corrections.

Response to the reviewers' comments

Response to Reviewer #1

General comments:

The authors have mostly answered my questions. I have a few additional comments. I think the finding that Zyxin is involved in fusion of WPBs is novel and interesting and well supported. The model of actin recruitment the author's propose, however, is still not completely supported by their data. In fact, their new added data shown in Supplementary Figure 5 now supports the more traditional model that new actin monomers are recruited from the cytosol and "coat" large fusing vesicles during exocytosis. This coating model is standard in the field and supported by many other papers involving lamellar body exocytosis in alveolar cells as well cortical granules in eggs and other exocytic systems. The new idea the authors present here is that the coating of WPBs occurs with the help of "pre-existing" actin filaments (abstract, page 4 line 2, page 6 line 3, page 11 line 2) that rearrange to (loosly???) surround vesicles before fusion. The authors try to distinguish what they are seeing from the standard model by renaming the actin structures an "actin framework".

We thank the reviewer for the positive comments.

Specific comments:

1. I would recommend that the authors take a more inclusive tone in their discussion and clearly state that the actin recruitment and coating they observe with high-resolution TIRF-SIM imaging is likely a combination of new recruitment of actin monomers from the cytosol and a reorganization of existing filaments which fits with existing models. There is no strong data to support that the process they observe is different from what has been observed and presented in the past.

We agree with the reviewer that the actin recruitment and coating appear to be a combination of new recruitment of actin monomers from the cytosol and a reorganization of existing filaments. Based on this model, we thus revised our discussion. (Page 11)

2. It might even help to move Supplemental Figure 5 to the main text and figures as this is a key experiment.

We are very pleased that the reviewer value the results of the Supplemental Figure 5 and removed it to the main text as Figure 2d and 2e.

2. These issues could be fixed by a careful re-working of their text with clear

interpretations of their data and a speculative cartoon model. Furthermore, the idea that the actin network is “strengthened”, pg. 6. Line 9 or “structurally loose” pg 7. Line 10 are mechanical arguments. The authors provide no direct experimental evidence for this in this paper. Please remove these arguments or discuss the signals as “denser---i.e. more fluorescent signal etc”. I mention this because something that is dispersed but rigid could be very strong and something dense and not networked could be very weak.

We are very grateful for these constructive comments. Following the reviewer’s suggestion, we revised our descriptions for the results accordingly, and discussions with a cartoon (Figure 8) for the combination model as mentioned above.

Response to Reviewer #2

Comments:

The authors have addressed the experimental concerns I raised. It should be noted that the results have significantly changed the interpretation of the results in that, in contrast to the assertions made in the original submission, it now appears that actin coating occurs via de novo assembly and that the reorganization of the cortical f-actin detected by TIRF-SIM does not reflect coating per se. This does not bother me too much as the reorganization itself is novel and it is possible that the preexisting cortical actin contributes indirectly to coat formation.

We thank the reviewer for the positive comments, and agree with the reviewer that the reorganization from the preexisting cortical actin contributes to coat formation, which was discussed with a newly added cartoon (Figure 8).

Response to Reviewer #3

General Comments:

This manuscript on the role of zyxin in von Willebrand factor (vWF) extrusion from endothelial cells highly novel using state-of-the-art imaging technology to understand better the process of exocytosis is novel and advances the field of regulated extrusion of granules. The imaging studies as far as I can interpret them support the conclusions. Also, the mice studies and biology studies of the released vWF are properly done and interpreted.

We thank the reviewer for the positive comments.

Major comments:

1. There is still not a full quantitation of data presented as images and videos. For example, In supplementary Fig 1, there is the claim that the Weibel Palade Bodies (WPB) lie along stress fibers by the number of cells examined and quantitation of that statement is missing. Similarly, other figures throughout the paper have biological statements made without a full quantitation. This is less an issue than the first submission, but still remains. Some of the now included quantitative data (e.g. Supplementary Figs 5B and 6B) are incompletely described methodologically or how the analysis was done.

We are grateful for the constructive comments and have revised the issues by adding the quantitative data (Figure 3d and Supplementary Figure 1) and methodological descriptions. (Method part in Page 18 and the legends)

2. Also lots of minor issues like inconsistent abbreviations and duplicated sentences are present. The supplemental videos need legends in the supplement section to explain what is being seen.

We corrected the mistakes and added the legends for the supplemental videos in the supplement section.

Minor Concerns:

1. While the manuscript focuses on WPB exocytosis, there are studies of platelet vWF and there is the low hanging fruit of whether vWF exocytosis in platelets is also zyxin dependent. People interested in vWF biology, separate from those interested in exocytosis would be interested in whether alpha granule release might also follow the same pathway and release of vWF following platelet activation might add to this manuscript.

We thank the reviewer for the insightful comments. We agree with the reviewer that it would be interested in whether VWF release from the alpha granules in platelets also follows the same pathway. Since the present study is focused on the role of zyxin in endothelial cells, we will systemically address this issue in the future study.

2. I will return an annotated version of the manuscript with many minor/moderate corrections.

We corrected the mistakes.